# Two novel, tightly linked, and rapidly evolving genes underlie *Aedes aegypti* mosquito reproductive resilience during drought

**Krithika Venkataraman**[1]\*, **Nadav Shai**[1,2], **Priyanka Lakhiani**[1,3], **Sarah Zylka**[1], **Jieqing Zhao**[1], **Margaret Herre**[1,4], **Joshua Zeng**[1], **Lauren A Neal**[1], **Henrik Molina**[5], **Li Zhao**[3], **Leslie B Vosshall**[1,2,4]\*

[1]Laboratory of Neurogenetics and Behavior, Rockefeller University, New York, United States; [2]Howard Hughes Medical Institute, New York, United States; [3]Laboratory of Evolutionary Genetics and Genomics, Rockefeller University, New York, United States; [4]Kavli Neural Systems Institute, New York, United States; [5]Proteomics Resource Center, Rockefeller University, New York, United States

**\*For correspondence:**
krithika.venkataraman@gmail.com (KV);
leslie@rockefeller.edu (LBV)

**Competing interest:** The authors declare that no competing interests exist.

**Abstract** Female *Aedes aegypti* mosquitoes impose a severe global public health burden as vectors of multiple viral pathogens. Under optimal environmental conditions, *Aedes aegypti* females have access to human hosts that provide blood proteins for egg development, conspecific males that provide sperm for fertilization, and freshwater that serves as an egg-laying substrate suitable for offspring survival. As global temperatures rise, *Aedes aegypti* females are faced with climate challenges like intense droughts and intermittent precipitation, which create unpredictable, suboptimal conditions for egg-laying. Here, we show that under drought-like conditions simulated in the laboratory, females retain mature eggs in their ovaries for extended periods, while maintaining the viability of these eggs until they can be laid in freshwater. Using transcriptomic and proteomic profiling of *Aedes aegypti* ovaries, we identify two previously uncharacterized genes named *tweedledee* and *tweedledum,* each encoding a small, secreted protein that both show ovary-enriched, temporally-restricted expression during egg retention. These genes are mosquito-specific, linked within a syntenic locus, and rapidly evolving under positive selection, raising the possibility that they serve an adaptive function. CRISPR-Cas9 deletion of both *tweedledee* and *tweedledum* demonstrates that they are specifically required for extended retention of viable eggs. These results highlight an elegant example of taxon-restricted genes at the heart of an important adaptation that equips *Aedes aegypti* females with 'insurance' to flexibly extend their reproductive schedule without losing reproductive capacity, thus allowing this species to exploit unpredictable habitats in a changing world.

## Editor's evaluation

This important study focuses on egg retention in the face of desiccating conditions in the mosquito *Aedes aegypti*. The research identifies genes associated with a trait that could be important to explain the vectorial capability of *Aedes aegypti* to transmit disease and expand into a cosmopolitan range. The presented evidence is convincing and the implications are well-articulated. The results should be of importance for ecological geneticists and vector biologists alike.

## Introduction

Extraordinary adaptations are ubiquitous across the animal kingdom within every habitat. Adaptations can be behavioral, physiological, or structural, and are evolutionarily selected to enable members of a species to persist by providing survival value. Ecosystems within which animals exist are in constant flux. When faced with changing habitats, animals must act flexibly and appropriately to survive and reproduce. For example, when the river-dwelling African lungfish (*Protopterus annectens*) experiences food and water scarcity during drought, it burrows into the dried riverbed, forming a cocoon with secreted mucus. It can survive for years while remaining metabolically dormant but within a week of rainfall, it reawakens and resumes normal metabolism (*Chng et al., 2017*; *Heimroth et al., 2018*). Among birds, species like the blackcap (*Sylvia atricapilla*) exhibit genetically encoded seasonal migratory behaviors that rapidly evolve in the face of changing resource availability, resulting in new migratory routes and destinations (*Berthold and Querner, 1981*; *Berthold et al., 1992*; *Delmore et al., 2020*). Land mammals such as marsupials alter the timing of their reproductive cycle in response to offspring-derived cues (*Renfree, 1979*). Tammar wallaby (*Macropus eugenii*) mothers can newly conceive while carrying previously birthed offspring in their pouches, but they developmentally arrest conceived embryos at the 100 cell stage, only resuming embryonic development after the pouch offspring has finished suckling and left to live independently (*Renfree, 1979*; *Tyndale-Biscoe et al., 1974*).

Mosquitoes that lay eggs at the edge of freshwater and go through an aquatic life cycle as larvae and pupae are very susceptible to fluctuating precipitation patterns and climate change-driven catastrophes like drought (*Gu et al., 2020*; *Cook et al., 2020*; *Hopp and Foley, 2003*; *Caldwell et al., 2021*). Despite climate variations, *Aedes aegypti* mosquitoes are highly invasive on almost every continent and pose a serious, immediate, and growing threat to global public health (*Brown et al., 2014*; *Juliano and Lounibos, 2005*; *Lounibos and Kramer, 2016*; *Kraemer et al., 2019*). While biting multiple humans to obtain the protein-rich blood they require to develop each batch of eggs, these mosquitoes have evolved as efficient vectors of arboviral infections such as yellow fever, Zika, dengue, and chikungunya, and of parasitic infections such as lymphatic filariasis (*WHO, 2017*). Domestic strains of *Aedes aegypti* prefer hunting and biting humans over other vertebrate hosts (*Harrington et al., 2001*; *Harrington et al., 2014*; *McBride et al., 2014*), and prefer laying eggs on moist surfaces proximal to freshwater in natural and manmade containers found around human settlements (*Bentley and Day, 1989*; *Matthews et al., 2019*). Female *Aedes aegypti* typically mate once in their lifetime (*Gwadz and Craig, 1968*), storing sperm in specialized organs called the spermathecae from which sperm are released to fertilize eggs post-ovulation, as eggs are in transit through the reproductive tract en route to being laid (*Jones and Wheeler, 1965*; *Degner and Harrington, 2016*). Once laid at the edge of freshwater, eggs darken and harden, and embryogenesis occurs within the eggshell (*Li, 1994*). After this, if conditions are suboptimal for hatching (*Rezende et al., 2008*), a developmental arrest state is triggered for up to 3–6 months to prevent embryo desiccation. Embryos then hatch when pools of freshwater become available again in their surroundings, and when aquatic larval and pupal development can be completed before eclosion to the terrestrial adult stage (*Clements, 1963a*).

The innate behaviors of an adult female *Aedes aegypti* mosquito are centered on the appropriate selection of a reproductive strategy that balances tradeoffs between internal energetic resources and external environmental conditions. Female mating, host-seeking, and egg-laying behaviors are inextricably linked, each proceeding only when the necessary 'checkpoints' have been cleared (*Clements, 1963b*). For example, females will not lay most, if any, of their eggs before they have mated (*Villarreal et al., 2018*). Females will suppress their attraction to hosts while eggs are developing and only restore attraction once eggs are laid (*Klowden and Lea, 1978*; *Klowden and Lea, 1979b*; *Klowden and Lea, 1979a*; *Klowden, 1994*; *Liesch et al., 2013*; *Duvall et al., 2019*). Females will not lay eggs both until and unless they locate freshwater, retaining them in their ovaries as needed (*Matthews et al., 2019*; *Judson, 1968*; *Day, 2016*). This interconnectedness of innate behaviors ensures that reproductive steps proceed in the order required for offspring survival. Because precise temporal control of egg-laying without loss of viability is an adaptation that maximizes the reproductive resilience and the fitness of *Aedes aegypti* females, understanding its basis will illustrate how this species is able to invade otherwise inhospitable ecological niches. Despite the importance of this question, little is known about how females are able to retain viable eggs in their ovaries during periods of prolonged drought.

Here we show that under drought-like conditions simulated in the laboratory, *Aedes aegypti* females will robustly retain eggs in their ovaries until freshwater is located. Under optimal conditions when freshwater is plentiful, females will lay eggs 3–4 days after a blood meal. We restricted access to freshwater for 4–12 days post-blood meal. A considerable proportion of eggs laid after extended retention to at least 12 days post-blood-meal were viable, hatching at high rates. We identified two previously uncharacterized, tightly linked genes – here named *tweedledee* and *tweedledum* for the curious pair of characters in Lewis Carroll's 1871 book, 'Through the Looking-Glass and What Alice Found There' – that are adult female-specific and ovary-enriched in their expression. The expression of these genes is dramatically upregulated in the ovaries only during the period in which females retain eggs, and the genes are spatially limited to cells that encapsulate mature eggs. Both genes are taxon-restricted, with no detectable orthology except in *Aedes albopictus,* a similarly invasive disease vector mosquito species that is~70 million years diverged from *Aedes aegypti* (***Chen et al., 2015***). In *Culex quinquefasciatus* and several *Anopheles* mosquito species, we identify putative orthologs with no sequence homology to *tweedledee* or *tweedledum*, but which bear other featural similarities such as synteny, gene structure, gene size, and the presence of signal peptides in the predicted proteins. *Aedes aegypti tweedledee* and *tweedledum,* as well as the *Anopheles gambiae putative ortholog*, are rapidly evolving genes within their respective species, and show strong signatures of positive selection. Using loss-of-function mutagenesis that disrupts both genes, we show that *Aedes aegypti tweedledee* and *tweedledum* are specifically required for extended retention of viable eggs under suboptimal drought conditions. Without *tweedledee* or *tweedledum,* mated, blood-fed *Aedes aegypti* females lose their reproductive 'insurance', such that when egg retention is triggered by restricted freshwater access due to drought-like conditions, most of the eggs they have matured no longer generate viable offspring if laid. Our results suggest that *tweedledee* and *tweedledum* play a crucial role in maintaining the reproductive resilience of female *Aedes aegypti* mosquitoes faced with fluctuating precipitation cycles and unpredictable drought-like conditions. This work thus illustrates a globally relevant example of rapidly evolving, taxon-restricted genes enabling an adaptation that allows *Aedes aegypti* mosquitoes to reproduce and thrive in otherwise inaccessible and inhospitable ecological niches.

## Results

### Female innate reproductive behaviors are interconnected

To ensure that the timing and sequence of steps in a female *Aedes aegypti* mosquito's reproductive cycle is appropriate for maximal reproductive output, the innate behaviors enabling access to blood meal sources, sperm, and freshwater egg-laying substrates are interconnected (***Figure 1A***). Under controlled laboratory conditions, consistent with previous results (***Klowden and Lea, 1978***; ***Klowden and Lea, 1979b***; ***Klowden and Lea, 1979a***; ***Klowden, 1994***; ***Liesch et al., 2013***; ***Duvall et al., 2019***), we showed that virgin and mated females were both attracted to human hosts prior to blood-feeding, and mated females showed significantly stronger levels of attraction (***Figure 1B***). After blood-feeding, mated females suppressed their drive to hunt during egg development (***Figure 1C–E***). Mated females continued to suppress their host-seeking drive when they were forced to retain mature eggs in their ovaries, as in the absence of an egg-laying substrate, for moderate (6 days post-blood-meal, ***Figure 1C***) or extended (12 days post-blood-meal, ***Figure 1D***) periods. Using individual egg-laying vials (***Figure 1—figure supplement 1A***), we found that ~80% of mated females 6 days post-blood-meal completed egg laying within 3 hr of gaining access to freshwater (***Figure 1—figure supplement 1B***). Females showed restored attraction to humans within 2 hr of completing egg laying (***Figure 1—figure supplement 1C–D***). This shows that attraction to humans is only fully restored upon completion of egg-laying, at which time the female is ready to initiate a second cycle of reproduction. The dynamics of mosquito attraction to humans is similar across multiple cycles of reproduction (***Figure 1E***), which suggests that attraction to humans – a de facto protein-feeding drive – is strongly dependent on reproductive physiology. Virgin females, in contrast to mated females, laid few melanized eggs when provided access to freshwater 6 days post-blood-meal (***Figure 1F***). Females that have both mated and have matured eggs within 3–4 days post-blood-meal must locate freshwater to lay their eggs (***Figure 1G***). If freshwater is not available, mated females will refrain from depositing their entire clutch of eggs (***Figure 1G***; ***Matthews et al., 2019***). Blood-feeding and mating

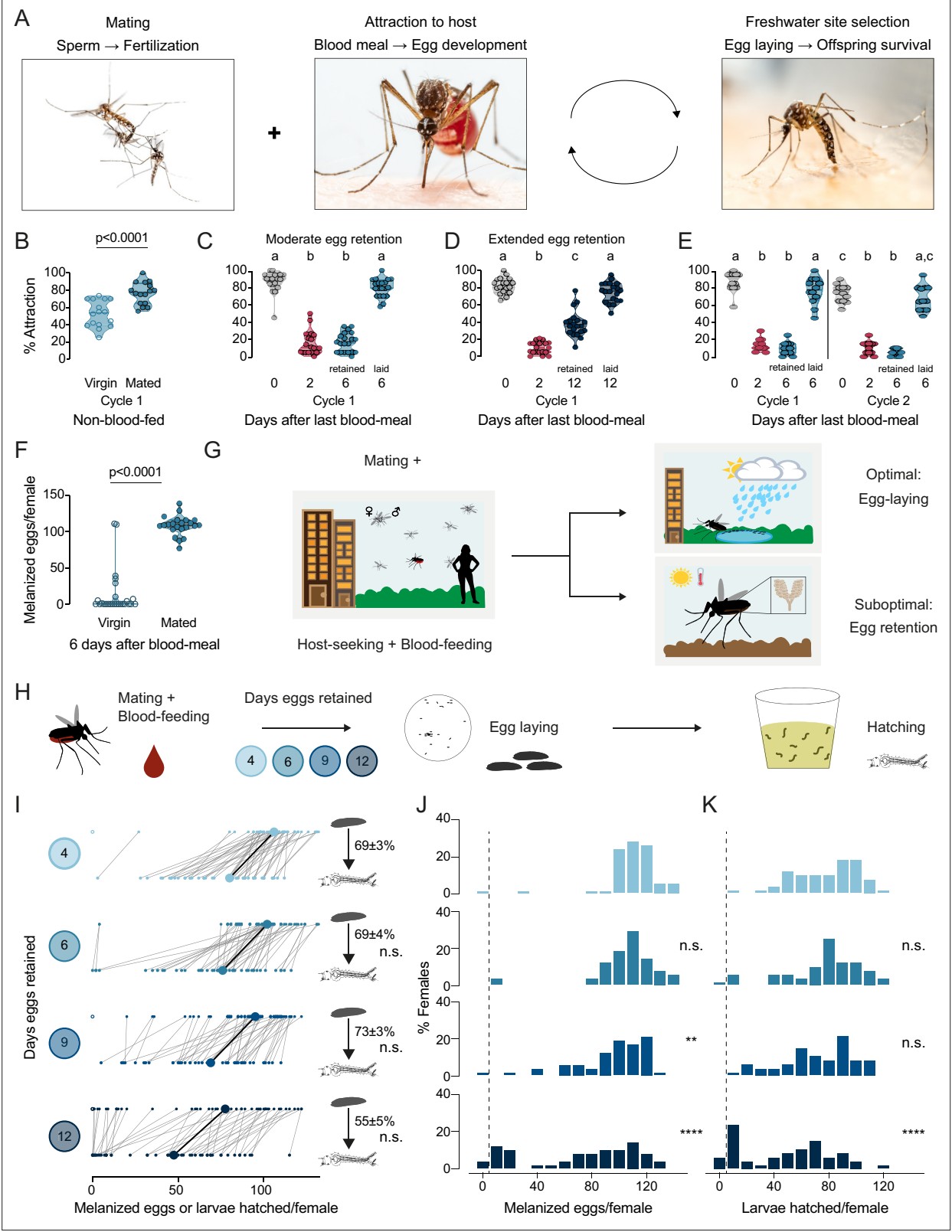

**Figure 1.** *Aedes aegypti* female reproduction is drought-resilient. (**A**) *Aedes aegypti* male and female mating (left), female blood-feeding from a human host (center), and female laying eggs in water (right). (**B–E**) Attraction of wild type females to a human forearm at the indicated reproductive state and cycle. Females are mated for all host-seeking experiments except where specified in (**B**). Each point represents a single trial with ~20 females, n=12–20 trials/group. Data are plotted as violin plots with median and 1st/3rd quartiles and showing all data points. Data labeled with different letters

*Figure 1 continued on next page*

*Figure 1 continued*

are significantly different: (B) Unpaired t-test, p<0.0001. (C) Kruskal-Wallis, Dunn's multiple comparisons test, p<0.05. (D, E) one-way ANOVA, Tukey's multiple comparisons test, p<0.05. (F) Number of melanized eggs laid by mated or virgin females. Data points from single females, n=22–26 females/group, shown as a violin plot with median and 1st/3rd quartiles with all data points. Mann-Whitney test, p<0.0001. (G) Schematic of a female mosquito's reproductive decision point after egg maturation under optimal and suboptimal egg-laying conditions of freshwater abundance and scarcity, respectively. (H) Schematic of experiment to test effect of egg retention on melanized egg laying and larval hatching. (I) Number of melanized eggs laid (top) and larvae hatched (bottom) by single females that experienced the indicated egg retention periods. Females laying no melanized eggs are depicted by open circles. Lines connect melanized eggs and hatched larvae from the same individual. Larger circles and bold lines represent medians. Numbers at right show hatching rate (mean ± S.E.M) from each egg retention group, n=46–50 females/group. (J, K) Distribution of melanized eggs (J) and larvae hatched (K) after the indicated length of egg retention, analyzed from data in (I). Zero values are binned separately for each group. All other bins are groups of 10 starting with [1-10] and with closed/inclusive intervals. The groups for number of melanized eggs (J), number of larvae hatched (K), and % hatched (I, numbers), respectively, at 6-, 9-, and 12 days post-blood-meal are compared to 4 days post-blood-meal to determine significant difference (Kruskal-Wallis, Dunn's multiple comparison test; n.s.: not significant, p>0.05; **p<0.01; ****p<0.0001). Mosquito photographs (A): Alex Wild.

The online version of this article includes the following figure supplement(s) for figure 1:

**Figure supplement 1.** Timing of egg-laying and return to human host seeking.

---

are thus decoupled – either one can occur first – but egg-laying, the ultimate step in a female's reproductive sequence, is tightly coupled to both mating and blood-feeding and requires both to occur.

## Drought induces extended retention of viable eggs

When a mated female mosquito has converted blood meal nutrients into mature eggs over 3–4 days, she must not only make the decision of where to lay her eggs, but she must also appropriately time her egg-laying decisions to ensure maximal offspring survival. We measured egg retention in the laboratory by simulating drought-like conditions of varying durations (*Figure 1H*). Females that engorged on a full blood meal laid ~100–110 melanized eggs at the edge of freshwater 4 days after a blood meal, of which ~70% hatched (*Figure 1I–K*). Even though the number of melanized eggs laid decreased with increasing length of egg retention, the proportion of viable eggs remained consistently high even after extended egg retention to at least 12 days post-blood-meal (*Figure 1I–K*). These results show that wild type *Aedes aegypti* mosquitoes demonstrate remarkable reproductive resilience during drought by retaining viable eggs until freshwater becomes available.

## *tweedledee* and *tweedledum* are ovary-enriched with temporally restricted expression

How female *Aedes aegypti* mosquitoes carrying mature eggs in their ovaries maintain the potential for subsequent fertilization, laying, and hatching of their eggs after different lengths of retention remains unexplored. To identify candidate genes regulating the retention of viable eggs in *Aedes aegypti* ovaries post-maturation, we used bulk RNA-sequencing (RNA-seq) to profile ovaries across 11 different time-points in their first cycle of reproduction (*Figure 2A*). Principal component analysis (PCA) of the ovary RNA-seq dataset shows that replicates within each reproductive stage cluster together, and that principal components 1 and 2 (PC1 and PC2) separate the reproductive stages from each other (*Figure 2B*). These data reflect the major transcriptional changes in the ovary across the reproductive cycle and highlight that each reproductive stage is distinctly and tightly regulated.

Ovaries carrying mature eggs occupy much of the female mosquito's abdomen, requiring redirection of her energy resources toward maintaining her eggs. Therefore, we hypothesized that candidate regulators of viable egg retention would be abundantly expressed across egg retention time-points, with specific upregulation at time-points when eggs are retained compared to pre-blood meal or during egg development. We expected that post-egg-laying, the expression of these genes would eventually decline as the female transitions out of her reproductive state. To identify putative candidates, we generated a list of genes ranked by abundance at the three time points when eggs were retained (*Figure 2C*). Among the top 50 most abundant genes, we identified two that showed striking regulation of expression across the reproductive cycle (*Figure 2D and E*). These previously uncharacterized genes, *LOC5563800* and *LOC5566109*, which we named *tweedledee* and *tweedledum* respectively, show similar and striking patterns of regulation in our transcriptomic dataset (*Figure 2D and E*). *tweedledee* is expressed in females before a blood meal and during egg

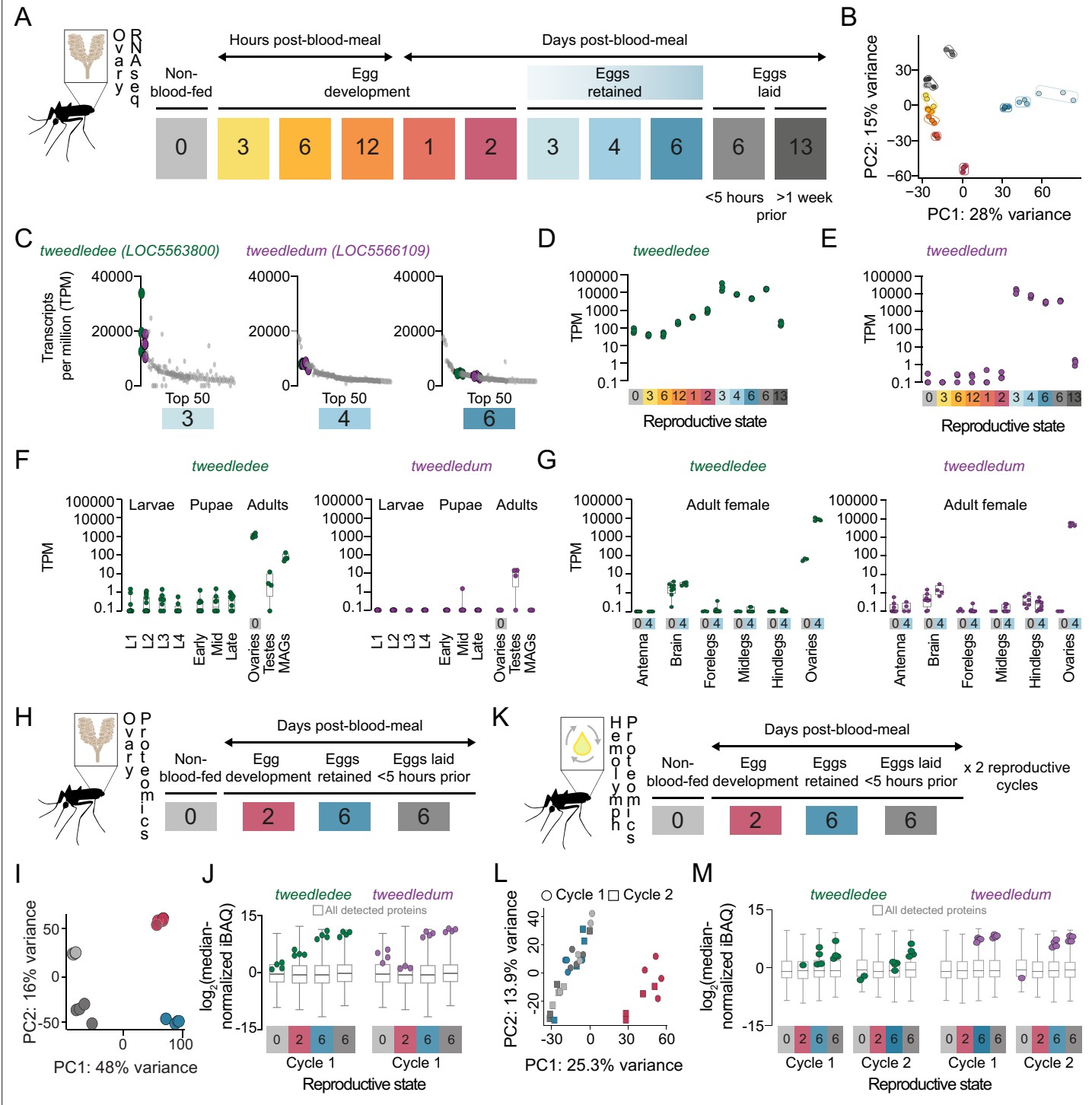

**Figure 2.** *tweedledee* and *tweedledum* are ovary-enriched and strongly upregulated during egg retention. (**A**) Reproductive time-points sampled for bulk ovary RNA-sequencing (RNA-seq), n=3 replicates/group, 11 groups. (**B**) Principal component analysis (PCA) of DESeq2-normalized, transformed counts from ovary RNA-seq. (**C**) Top 50 most abundant transcripts ranked by median transcripts per million (TPM) for egg retention groups, 3-, 4-, and 6 days post-blood-meal (total = 150 transcripts). Gray dots represent replicates for each transcript in the top 50, green dots indicate *tweedledee*, and purple dots represent *tweedledum*. (**D–E**) Transcript expression pattern in the ovaries of *tweedledee* (**D**) and *tweedledum* (**E**). The blue rectangle indicates the period of egg retention. (**F**) TPM values for *tweedledee* (left) and *tweedledum* (right) during larval, pupal, and adult stages of development [data from ***Matthews et al., 2018***], MAGs = male accessory glands, n=4–13 replicates/group. (**G**) TPM values for *tweedledee* (left) and *tweedledum* (right) in adult female tissues (data originally from ***Matthews et al., 2016***), reanalyzed in ***Matthews et al., 2018***, n=3–8 replicates/group. (**H**) Reproductive time-points sampled for ovary proteomics, n=4 replicates/group, 4 groups. (**I**) PCA of iBAQ values from ovary proteomics.

*Figure 2 continued on next page*

*Figure 2 continued*

(**J**) Distribution of iBAQ values as a metric of abundance for all proteins detected per group in ovary proteomics. Overlaid green dots represent individual replicate values for tweedledee and purple dots represent replicates for tweedledum. All values are pre-imputation and represent log$_2$-transformed median iBAQ signals normalized by subtracting the median iBAQ signal for the group. (**K**) Reproductive time-points sampled for hemolymph proteomics, n=4 replicates/group, 8 groups. (**L**) PCA of iBAQ values from hemolymph proteomics. (**M**) Distribution of iBAQ values as a metric of abundance for all proteins detected per group in hemolymph proteomics. Overlaid green dots represent individual replicate values for tweedledee and purple dots represent replicates for tweedledum. All values are pre-imputation and represent log$_2$-transformed median iBAQ signals normalized by subtracting the median iBAQ signal for the group. Box plots in F, G, J, M: median, 1st/3rd quartile, minimum to maximum whiskers.

development, but its expression increases ~3 orders of magnitude during periods of egg retention (*Figure 2D*). The regulation of *tweedledum* is even more remarkable. It is present at less than 1 transcript per million (TPM) at non-blood-fed and egg production stages but rises to ~10,000 TPM during egg retention (*Figure 2E*).

Using published transcriptomes of developmental stages (*Matthews et al., 2018*) and adult tissues (*Matthews et al., 2016*) from *Aedes aegypti* mosquitoes, we discovered that *tweedledee* and *tweedledum* are adult-specific and female-enriched (*Figure 2F*). Both genes show ovary-enriched expression with strong upregulation post-egg maturation in the females (*Figure 2G*), with some expression in male reproductive tissues (*Figure 2F*). In addition to specific upregulation during egg retention, *tweedledee* shows basal, constitutive expression across several conditions and tissues, whereas the spatiotemporal expression of *tweedledum* is more tightly restricted (*Figure 2D–G*). These data in adult females show exquisite specificity of *tweedledee* and *tweedledum* expression in ovaries bearing mature eggs, strengthening the possibility that the genes are candidate regulators of viable egg retention.

We next performed quantitative proteomics profiling of the female ovaries across a subset of reproductive time-points corresponding to non-blood-fed, egg development, egg retention, and post-egg-laying states (*Figure 2H*). PCA shows that replicates within each stage again clustered together as in the RNA-seq dataset, and all reproductive states formed distinct clusters in PC1 and PC2, reflective of the ovaries being tightly and distinctly controlled across these different reproductive states (*Figure 2I*). Both tweedledee and tweedledum proteins were notably upregulated at the egg retention phase of the reproductive cycle (*Figure 2J*). Because tweedledee and tweedledum expression levels remain high in the ovaries when sampled <5 hr post-egg-laying when all mature eggs have been laid, we speculate that these genes are expressed in somatic tissues in the ovary (*Figure 2J*).

*tweedledee* and *tweedledum* are predicted, based on their sequence, to encode proteins with N-terminal signal peptides. To test if they are secreted, we profiled the proteome of the circulating hemolymph, the insect equivalent of blood. We collected hemolymph samples across non-blood-fed, egg development, egg retention, and post-egg-laying states in the first and second cycles of reproduction (*Figure 2K*). The hemolymph is in close apposition to the ovaries, and its contents during distinct reproductive time-points reflect interorgan communication (*Anderson and Spielman, 1971*; *Sun et al., 2000*; *Hansen et al., 2014*). PCA of the hemolymph proteome showed that at 2 days post-blood-meal, the composition of the circulating fluid is most distinct (separated by PC1) from other profiled time-points (*Figure 2L*). These findings are consistent with our expectations, as this is the only time-point profiled during which eggs are likely to still be maturing and during which the hemolymph is therefore transporting components for egg maturation (*Hagedorn, 1974*; *Hagedorn and Fallon, 1973*). Notable examples of hemolymph-transported proteins include the vitellogenins (yolk protein precursors), which we detect in our dataset (https://doi.org/10.5281/zenodo.5945524). These proteins are synthesized in the fat body, an analog of the vertebrate liver, and transported via the hemolymph to the ovaries where they are packaged into maturing eggs (*Hagedorn, 1974*; *Fallon et al., 1974*). We detected tweedledee and tweedledum protein in the hemolymph and found that they were both strongly upregulated in each of the reproductive cycles during egg retention and within 5 hr of egg-laying compared to pre-blood-meal, during egg development, or >1 week post-egg-laying (*Figure 2M*). These data together suggest that somatic ovary cells secrete tweedledee and tweedledum, and that their expression and secretion into the circulating hemolymph are both tightly regulated.

## *tweedledee* and *tweedledum* are expressed in cells encapsulating mature eggs

Within the ovaries, mature eggs are housed in individual chambers/follicles, encapsulated within a membrane of follicular epithelial cells (*Parks and Larsen, 1965*; *Bertram, 1959*). At the point of egg-laying, mature eggs transit out of their individual chambers and enter the calyx, a continuous tube through the center of the ovary connected to the oviducts (*Bertram, 1959*; *Curtin and Jones, 1961*). Eggs transit into the oviducts and are fertilized in the reproductive tract by sperm released from sperm storage organs, the spermathecae, before being ejected through the ovipositor (*Degner and Harrington, 2016*; *Bertram, 1959*; *Curtin and Jones, 1961*).

Using whole-mount ovary fluorescence RNA in situ hybridization we show that *tweedledee,* but not *tweedledum* transcripts are detectable in non-blood-fed ovaries (*Figure 3A*). *tweedledee* expression in non-blood-fed ovaries is restricted to calyx cells, and it is markedly absent in the primary follicles (*Figure 3A*). The primary follicles are comprised of seven nurse cells and an oocyte surrounded by somatic follicular epithelial cells (*Valzania et al., 2019*). Once the female consumes a blood meal, the primary follicle develops into an egg, with the surrounding follicular epithelial cells secreting eggshell proteins and other components onto it (*Valzania et al., 2019*; *Isoe et al., 2019*). The oocyte is characteristically marked by *vitellogenin receptor* (*LOC5569465*) expression (*Figure 3A, B, D and F*). The *vitellogenin receptor* gene enables receptor-mediated endocytosis of yolk precursor proteins into the egg after a blood meal (*Sappington et al., 1996*).

Because of technical limitations of performing fluorescence RNA in situ hybridization on intact ovaries both during egg development and during retention of fully mature eggs due to optical opacity of the ovary and difficulties with probe penetration, we utilized ovaries 6 days post-blood-meal within 5 hr of egg-laying to identify the cells expressing *tweedledee* and *tweedledum* (*Figure 3B–E*). Since ovary RNA-seq data suggest both *tweedledee* and *tweedledum* transcripts are abundantly expressed <5 hr post-egg-laying (*Figure 2D and E*), we postulated that this time-point would allow us to identify which cells express *tweedledee* and *tweedledum*. Ovaries collected within 5 hr of egg-laying have two different types of egg follicles (*Figure 3B–E*): the remnants of primary follicles which held mature eggs prior to laying and a secondary follicle that was previously attached to the primary follicle, and that is ready to develop into a new egg upon consumption of a second blood meal (*Bertram, 1959*; *Valzania et al., 2019*; *Riehle and Brown, 2002*). *tweedledee* was detected in the calyx through which eggs transit (*Figure 3B and C*) as well as in the follicular epithelial cells of the primary follicle remnants (*Figure 3B, C and E*). *tweedledum* was also expressed in the follicular epithelial cells of primary follicle remnants, and solely in these cells together with *tweedledee* (*Figure 3B–E*). Notably, neither of the transcripts were expressed in secondary follicles (*Figure 3B–E*).

We additionally examined *tweedledee* and *tweedledum* expression >1 week post-egg-laying when the gross morphology of the ovary resets and bears closer resemblance overall to non-blood-fed ovaries. At this time-point, *tweedledum* expression was once again undetectable, and *tweedledee* was exclusively expressed in the calyx (*Figure 3F*). The patterns of *tweedledee* and *tweedledum* expression detected using fluorescence RNA in situ hybridization in non-blood-fed ovaries (*Figure 3A*) and in ovaries <5 hr (*Figure 3B–E*) or >1 week post-egg-laying (*Figure 3F*) validate the expression patterns from the respective time-points in the ovary RNA-seq data (*Figure 2F*). Overall, the robust expression of *tweedledee* and *tweedledum* in the follicular epithelial cells of primary follicle remnants and the added expression of *tweedledee* in the calyx suggests that these genes, either independently or together, are poised to play a role in protecting eggs specifically during egg retention and while they are *en route* to being laid.

## *tweedledee* and *tweedledum* are linked and taxon-restricted

We next turned to the *Aedes aegypti* genome for clues on the function and evolutionary origin of *tweedledee* and *tweedledum*. The genes are located next to each other on chromosome 2, and both have a short first exon and a longer second exon (*Figure 4A*). These genes are predicted to encode small proteins (tweedledee: 216 amino acids; tweedledum: 116 amino acids), both with N-terminal signal peptides and disordered regions but no other known domains (*Figure 4B*). Although similar in many respects, the two genes and their encoded proteins bear no sequence similarity to each other. We calculated the guanine + cytosine (GC) content for all protein-coding genes in the *Aedes aegypti* genome. This metric is indicative of gene and transcript thermal stability, and thus also an

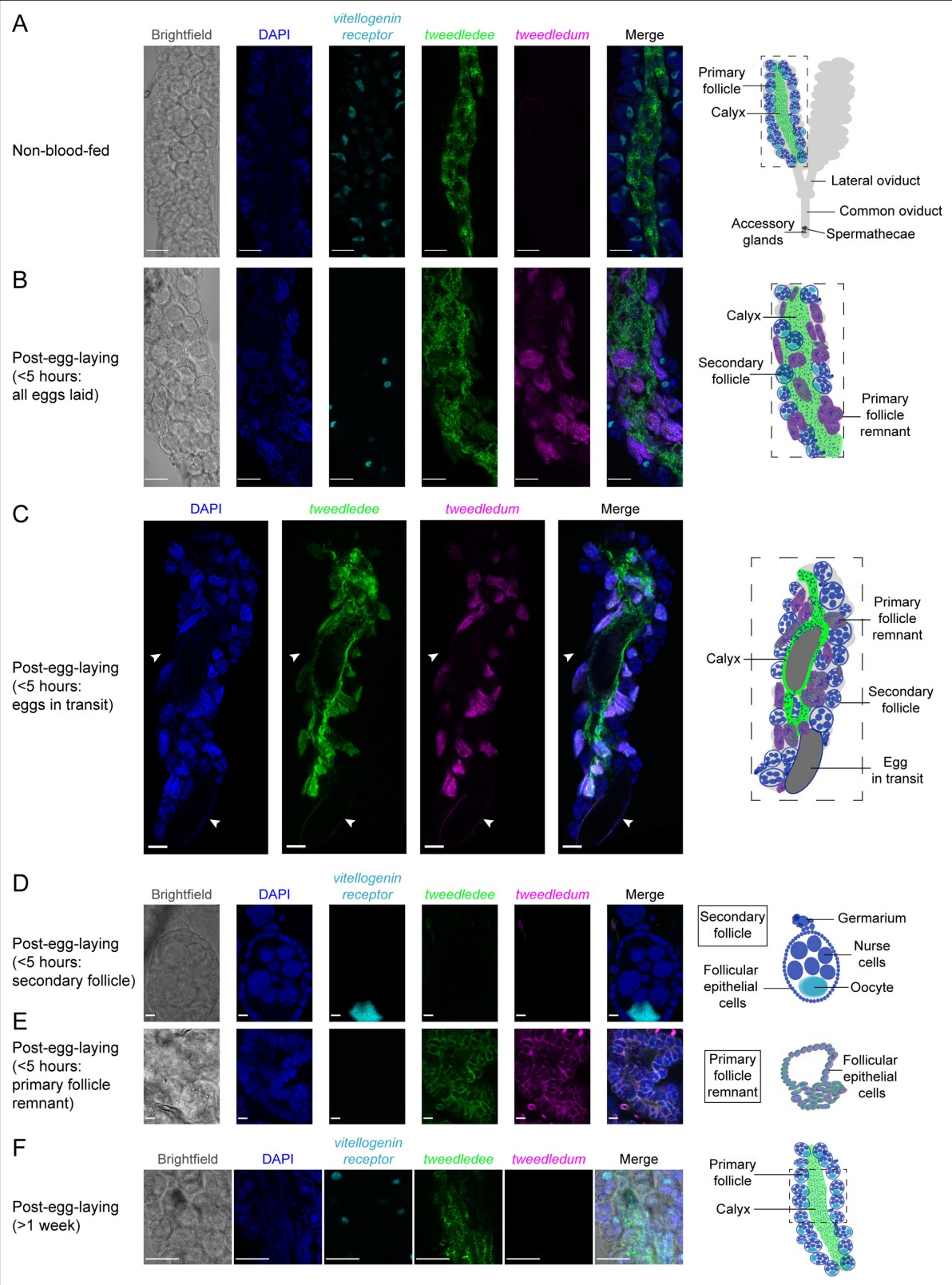

**Figure 3.** *tweedledee* and *tweedledum* are expressed in cells encapsulating mature eggs. (**A**) Left: Single confocal section of whole-mount fluorescence RNA in situ hybridization of a non-blood-fed ovary with the indicated probes. Right: cartoon of a pair of ovaries and the female reproductive system, with left ovary representing a cross-section of one ovary. (**B, C**) Left: Single confocal section of whole-mount fluorescence RNA in situ hybridization of an ovary <5 hr post-egg-laying with all eggs laid (**B**) or with two eggs in transit indicated by white arrows (**C**) with the indicated probes. Right:

*Figure 3 continued on next page*

*Figure 3 continued*

cartoons representing cross-section through a post-egg-laying ovary with probe expression patterns depicted in different ovary structures. (**D, E**) Left: Single confocal section of whole-mount fluorescence RNA in situ hybridization of an ovary <5 hr post-egg-laying with the indicated probes showing a secondary follicle ready to develop into an egg upon consumption of a second blood meal (**D**) and a post-egg-laying follicle that is the remnant of a primary follicle which previously contained an egg (**E**). Right: cartoons depicting *tweedledee* and *tweedledum* expression pattern uniquely in the post-egg-laying follicle/primary follicle remnant (**E**), but not in the secondary follicle expressing *vitellogenin receptor* (**D**). (**F**) Left: Single confocal section of whole-mount fluorescence RNA in situ hybridization of an ovary >1 week post-egg-laying with the indicated probes. Right: cartoon of a single ovary cross-section. Scale bars: 100μm in A-C, F and 10 μm in D-E.

important determinant shaping interactions between a species and its environment (*Šmarda et al., 2014*). Compared to all protein-coding genes, *tweedledee* (50% GC) and *tweedledum* (48% GC) fall in the 94th and 89th percentile, respectively (*Figure 4C*). Within the distribution of protein-coding genes containing a predicted signal peptide, the percentiles for *tweedledee* and *tweedledum* remain similar at 95th and 88th, respectively (*Figure 4C*). We next calculated the proportion of each amino acid residue in tweedledee and tweedledum and compared it to the average proportion of each amino acid residue across all proteins in the *Aedes aegypti* genome that contain a predicted signal peptide (*Figure 4D*). In all cases, we performed comparisons on proteins in their functional secreted form, with signal peptides cleaved in silico. Both tweedledee and tweedledum share compositional biases with each other relative to other secreted proteins encoded by the *Aedes aegypti* genome. They both show an underrepresentation of leucine, threonine, and glycine, an overrepresentation of aspartate, glutamate, alanine, valine, and serine, and entirely lack cysteine, tyrosine, and tryptophan (*Figure 4D*).

To explore the evolutionary history and origin of these genes, we searched for putative homologs. Using BLASTp, the only orthologs identifiable in Genbank at the time of analysis for both *tweedledee* and *tweedledum* with E-values <0.05 are in *Aedes albopictus,* another invasive mosquito vector ~70 million years diverged from *Aedes aegypti* (*Chen et al., 2015*). In both *Aedes aegypti* and *Aedes albopictus, tweedledee* and *tweedledum* or their respective orthologs are flanked by two conserved genes, *peritrophin-like* and *scratch* (annotated as *escargot* in *Aedes aegypti*) (*Figure 4E*). Using *peritrophin-like* and *scratch* as 'anchor' genes, we searched for other syntenic loci potentially containing *tweedledee* or *tweedledum* homologs in other mosquito species (*Figure 4E*). We found syntenic loci in several other mosquitoes, but not in any non-mosquito species, including *Drosophila melanogaster* flies. The *Drosophila melanogaster scratch* gene is located on chromosome 3 L and is not in close proximity to any *peritrophin-like* genes. There are several genes adjacent to *Drosophila melanogaster scratch*, but none have the gene or protein structure of *tweedledee* and *tweedledum*. In *Culex quinquefasciatus, Anopheles gambiae,* and several other *Anopheles* mosquito species both within and outside of the *Anopheles gambiae* complex, there are syntenic loci with conserved *peritrophin-like* and *scratch* genes (*Figure 4E*). In these *Culex* and *Anopheles* cases, the conserved genes flank a single uncharacterized gene that we hypothesize is an ancient putative ortholog of either *tweedledee* or *tweedledum* (*Figure 4E*). These genes bear no sequence homology between each other in *Culex* and *Anopheles* or to *tweedledee* or *tweedledum* in *Aedes,* but they are like *tweedledee* and *tweedledum* in many other aspects. First, they are all two exons long, with a short first exon and a longer second exon. Second, they are predicted to encode proteins of similar length ranging between 190 and 269 amino acids, and third, they are predicted to contain N-terminal signal peptides. Ordered by the topology of the mosquito phylogenetic tree, the protein sequences of tweedledee and tweedledum in *Aedes* or of the putative orthologs in *Anopheles* diverge more rapidly than the protein sequences of their flanking anchor genes within their respective genera (*Figure 4E*). In comparing the amino acid content of all putative orthologs (with signal peptides cleaved) to each other and to the *Aedes* tweedledee and tweedledum, we observed several similarities despite the rapid protein sequence divergence: all genes have no or very few cysteine or tryptophan residues, and an overrepresentation of glutamate and alanine (*Figure 4F*). Exonic sequences of *tweedledee*, *tweedledum*, and the *putative orthologs* in *Culex* and *Anopheles* also show strong similarities in the relative locations of their signal peptides and disordered domains as predicted by SignalP and IUPred2A, respectively (*Figure 4G*).

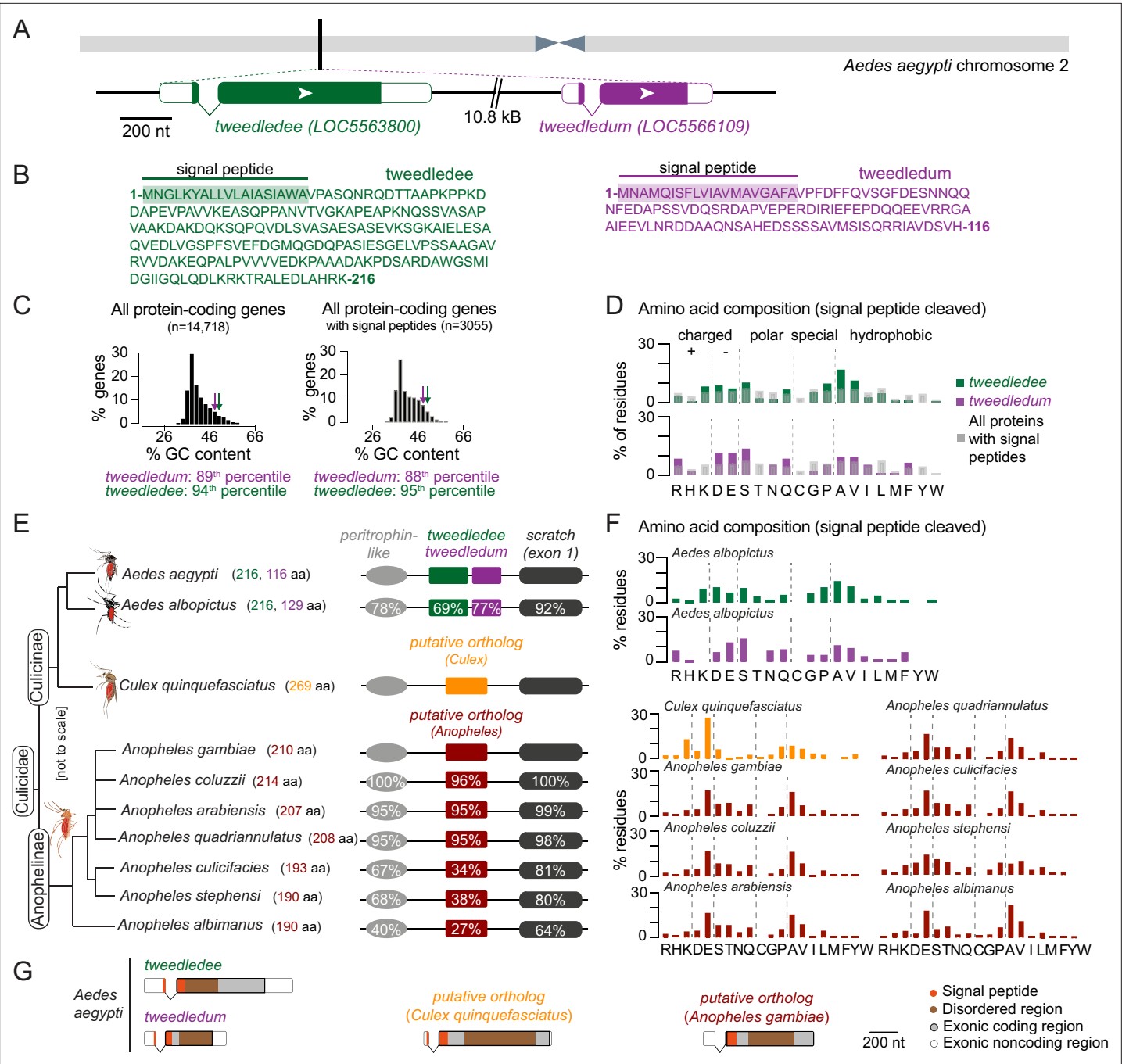

**Figure 4.** *tweedledee* and *tweedledum* are linked, taxon-restricted, syntenic, and rapidly evolving genes. (**A**) Chromosomal location and gene structure of *tweedledee* and *tweedledum*. (**B**) Amino acid sequences of tweedledee (left) and tweedledum (right) with predicted N-terminal signal peptides indicated. (**C**) GC content of all protein-coding genes in the *Aedes aegypti* (AaegL5) genome (left) and of all protein-coding genes with predicted signal peptides (right), with *tweedledee* and *tweedledum* indicated by arrows. (**D**) Amino acid composition of *Aedes aegypti* tweedledee and tweedledum, as compared to mean percent residue for 3,040 proteins with predicted signal peptides in the *Aedes aegypti* genome, calculated after signal peptide cleavage. (**E**) Syntenic loci in *Aedes*, *Culex*, and *Anopheles* mosquito species are shown (not to scale), ordered by the topology of the mosquito phylogenetic tree. The protein length of tweedledee, tweedledum, or the putative ortholog is shown in parentheses next to the species name. Protein sequence identity is shown for each gene as calculated using a reference species for each genus, either *Aedes aegypti* or *Anopheles gambiae*. For *scratch*, protein sequence identity was calculated by aligning exon 1 of each species due to a fragmented annotation in multiple reference genomes (see Methods). Accession numbers for all genes are at https://doi.org/10.5281/zenodo.5945524. (**F**) Amino acid composition of tweedledee and tweedledum in *Aedes albopictus* and of the putative ortholog in *Culex* and *Anopheles* species. (**G**) Gene structures of *Aedes aegypti tweedledee* and *tweedledum* and the putative ortholog in *Culex quinquefasciatus* and *Anopheles gambiae* are shown to scale with signal peptide and disordered domains annotated. The 3′UTR of the *Anopheles gambiae* putative ortholog is lacking in the current genome annotation.

# Rapid evolution of *tweedledee* and *tweedledum* is associated with drought-related environmental parameters

To assess whether the molecular evolution of *Aedes aegypti tweedledee* and *tweedledum* relative to the outgroup, *Aedes albopictus*, and of the *Anopheles gambiae putative ortholog* relative to the outgroup, *Anopheles stephensi*, is adaptive, we computed the ratio of non-synonymous (dN, amino acid-altering) to synonymous (dS, silent) mutations at each site (*Yang and Bielawski, 2000*). By calculating the distribution of dN/dS values for all protein-coding genes with unique outgroup orthologs in the *Aedes aegypti* and *Anopheles gambiae* genomes, we found that *tweedledee*, *tweedledum*, and the *putative ortholog* are in the 98th, 92nd, and 99th percentile, respectively (*Figure 5A*). When comparing the dN/dS of *tweedledee* and *tweedledum* to ovary-expressed genes, they are in 98th and 94th percentile. This suggests that compared to most protein-coding genes in mosquitoes, amino acid-altering mutations are more likely to reach fixation for *tweedledee*, *tweedledum*, and the *Anopheles putative ortholog*. A sliding-window analysis of dN/dS values across the coding sequences of *Aedes aegypti tweedledee* and *tweedledum* revealed that these high gene-wide dN/dS values are likely driven by rapid sequence divergence in specific regions around the middle of the gene (*Figure 5B*).

To investigate whether *tweedledee* and *tweedledum* evolved under positive selection, we turned to a large population genetic dataset (*Rose et al., 2020*) and performed McDonald-Kreitman (MK) tests (*McDonald and Kreitman, 1991*; *Begun et al., 2007*). The alpha values, a metric to describe the proportion of substitutions driven by positive selection, for *tweedledee* and *tweedledum* were 0.48 and 0.87, respectively (*Smith and Eyre-Walker, 2002*). The p-values for the MK test calculated using a Fisher's Exact test were 0.068 and 0.048 respectively, suggesting a high probability that the genes are evolving under positive selection (*Figure 5A*). For the *putative ortholog* in *Anopheles gambiae,* we used population data from the Ag1000G project (*The Anopheles gambiae 1000 Genomes Consortium, 2021*) to run the MK test. The alpha value and p-value were 0.76 and 0.0066, respectively (*Figure 5A*). These analyses together suggest that *tweedledee, tweedledum*, and the *putative orthologs* likely shared a common ancestor, and that the genes are diverging rapidly under strong selective pressure across the mosquito phylogeny.

Wild *Aedes aegypti* strains recently collected across sub-Saharan Africa for genome sequencing (*Rose et al., 2020*) inhabit diverse climates. If climate conditions impose selective pressure at the *tweedledee* and *tweedledum* locus, we may expect that in different strains, genomic signatures in this locus correlate with ecological parameters of their respective geographic regions. To address this question, we used population genetics data collected from 25 of the 27 sampling locations across sub-Saharan Africa (*Rose et al., 2020*). These populations of *Aedes aegypti* originated from regions with a range of ecological conditions, including highly seasonal, semi-arid climates where reproductive resilience achieved through flexible, robust egg retention in the female ovary could confer a marked fitness advantage. Using genomic sequences from 407 individuals across 25 populations of *Aedes aegypti* (*Rose et al., 2020*), we called variants for all 14,438 protein-coding genes, and correlated the alternate allele frequency in each population with 14 ecological variables, which comprise 2 anthropological variables: human population density and host-seeking preference index (*Rose et al., 2020*), and 12 Bioclimatic variables (*Fick and Hijmans, 2017*). To test each *Aedes aegypti* protein-coding gene for correlation to all 14 ecological variables, we computed a correlation test metric by combining correlations from each individual alternate allele and correcting for multiple testing (*Figure 5C*). When ranking all 14,438 protein-coding genes in *Aedes aegypti* according to our metric, we found that both *tweedledee* and *tweedledum* are in the top 90th percentile of genes with non-spurious correlations to ecological variables reflecting climate variability, such as mean diurnal range, temperature seasonality, and precipitation seasonality (*Figure 5C*). We additionally tested for statistical significance by performing independent permutation tests to correct for the gene sizes of *tweedledee* and *tweedledum*, respectively. Both *tweedledee* (*Figure 5D*) and *tweedledum* (*Figure 5E*) each showed significant correlation to multiple climate variables when compared to a distribution of 10,000 simulated genes with randomly-sampled genetic variants (p<0.05).

We verified this analysis method using *Or4*, a rapidly evolving olfactory receptor in *Aedes aegypti* that has been found to drive evolution to human preference via missense alleles that affect protein function (*McBride et al., 2014*; *Rose et al., 2020*). Using our analysis, we found that *Or4* is significantly correlated with human-related and other ecological variables that support findings from *Rose et al., 2020* (*Rose et al., 2020*; *Figure 5—figure supplement 1*). These strong correlations suggest

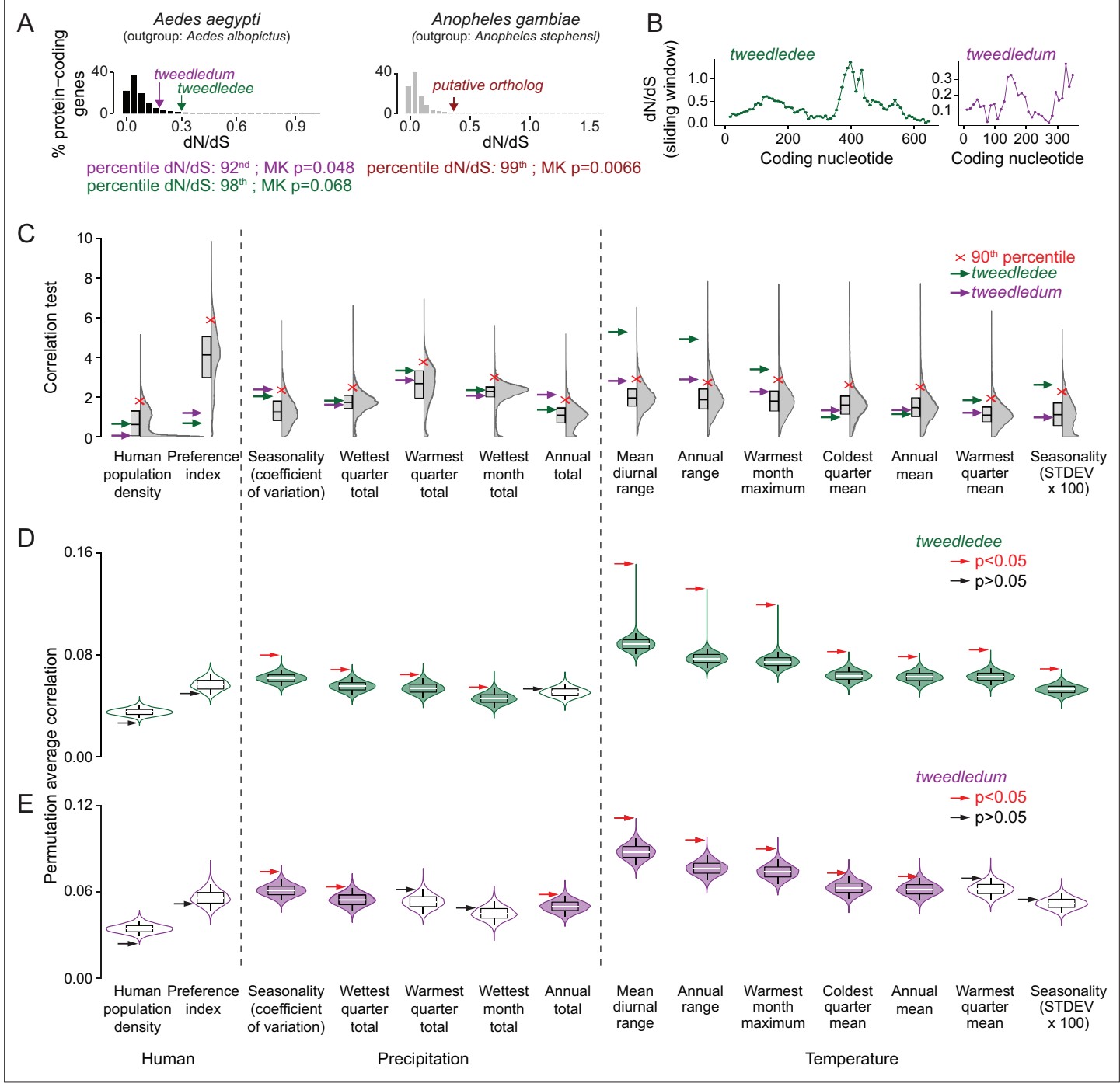

**Figure 5.** *tweedledee* and *tweedledum* are rapidly evolving and may be subject to climate variability-driven selective pressures. (**A**) The distribution of dN/dS values for 8,030 protein-coding genes in *Aedes aegypti* (outgroup: *Aedes albopictus*) and 9,958 protein-coding genes in *Anopheles gambiae* (outgroup: *Anopheles stephensi*). *tweedledee*, *tweedledum* and the *Anopheles gambiae* putative ortholog are shown with arrows. (**B**) dN/dS values were calculated for a 102-nucleotide sliding window size of 34 nucleotides each with a 3 amino acid overlap across the coding sequence of *Aedes aegypti tweedledee* and *tweedledum*. Coding sequences were aligned to orthologs in *Aedes albopictus*. (**C**) Correlation test for 12 bioclimatic and 2 anthropological variables (total=14 ecological variables) calculated using all genetic variants. The distributions of 14,438 protein-coding genes in *Aedes aegypti* are plotted with Correlation Test calculated and plotted on the y-axis using the following formula: -log$_{10}$(harmonic mean combined Pearson's correlation p-value). Individual Pearson's correlation for all genetic variants in all protein-coding genes were included in the calculation. Boxes show median and 1st/3rd quartiles. Red X indicates 90th percentile. Violin tails extend to show the entire range of data points. The positions of *tweedledee* (green) and *tweedledum* (purple) are indicated by arrows. (**D, E**) Permutation tests were conducted for *tweedledee* (**D**) and *tweedledum* (**E**). Genetic variants were randomly sampled from all protein-coding genes to simulate 10,000 genes with the same number of genetic variants as either *tweedledee*

*Figure 5 continued on next page*

Figure 5 continued

(358) or *tweedledum* (292). Boxes show median and 1st/3rd quartiles and whiskers extend to the 5th/95th percentiles. Significance was measured using p<0.05 (above the 95th percentile). Violin tails extend to show the entire range of data points. The positions of *tweedledee* (**D**) and *tweedledum* (**E**) are indicated by an arrow. Significance is indicated by arrow color (red: p<0.05; black: p>0.05). Violins with p<0.05 are shaded green for *tweedledee* (**D**) or purple for *tweedledum* (**E**).

The online version of this article includes the following figure supplement(s) for figure 5:

**Figure supplement 1.** *Or4* is significantly correlated with human-related and other ecological variables.

that segregating genetic variants in *tweedledee* and *tweedledum* may reflect adaptation in wild *Aedes aegypti* populations to local climate parameters, especially fluctuating temperature and precipitation – two ecological variables that are strongly linked (*Rose et al., 2020*).

## *tweedledee* and *tweedledum* are required for retention of viable eggs during drought

Under fluctuating climate conditions of intermittent precipitation, retaining viable eggs for extended durations may be an adaptive reproductive strategy for *Aedes aegypti* females. To test whether *tweedledee* and *tweedledum* are required under drought-like conditions for females to retain viable eggs for extended periods after blood feeding, we used CRISPR-Cas9 to generate mosquitoes with a large deletion at the *tweedledee* and *tweedledum* locus, here referred to as *Δdeedum* double mutants (*Figure 6A*, *Figure 6—figure supplement 1A*, *Figure 6—figure supplement 1—source data 1*). The 11.7 kb deletion starts within *tweedledee* exon 2 and ends in exon 2 of *tweedledum* (*Figure 6A*). The gene fusion resulting from the large deletion and several additional indels is predicted, in silico, to encode a protein with amino acids 1–53 of *tweedledee* conserved before the breakpoint junction, following which a frameshift is introduced that leads to fusion with 68 missense amino acids before a stop codon (*Figure 6A*). This deletion event in *Δdeedum* is predicted to produce null mutations in both *tweedledee* and *tweedledum*.

To characterize the reproductive behaviors of *Δdeedum* double mutant females compared to wild type females, we mated them to sibling males of their respective genotypes. To assess the general health of females, we tested their level of attraction to human hosts (*Figure 6B*) and their blood meal consumption (*Figure 6C*). Like wild type females, *Δdeedum* double mutant females were strongly attracted to a live human arm in a single stimulus olfactometer assay (*Figure 6B*). They approximately doubled their body weight from engorging on a blood meal (*Figure 6C*), and when presented with the same live arm stimulus 6 days following the blood meal while retaining eggs, they showed a suppressed host-seeking drive like wild type females (*Figure 6B*). Both wild type and *Δdeedum* females restored attraction to human hosts by 6 days after the blood meal if they had been provided freshwater to lay eggs 3–5 days after the blood meal (*Figure 6B*). Together, these host-seeking and blood-feeding results suggest that *Δdeedum* mutants are healthy, and that loss of *tweedledee* and *tweedledum* together does not affect attraction to human hosts, modulation of attraction following a blood meal and following egg-laying, or the ability to engorge on a full blood meal – all crucial behavioral checkpoints for reproductive success.

We next asked whether *Δdeedum* double mutant females have morphologically healthy ovaries and spermathecae with visually normal eggs and sperm, respectively. *Δdeedum* females that consumed a full blood meal developed mature eggs and retained them for at least 12 days after the blood meal in their ovaries. There were no grossly observable morphological defects in *Δdeedum* eggs or ovaries compared to wild type when ovaries were dissected and photographed 6 days (*Figure 6D*) or 12 days (*Figure 6G*) after the blood meal. Spermathecae, the organs specialized for sperm storage following a single mating event contained sperm that appeared motile in both wild type and *Δdeedum* mutants at both time-points (*Figure 6D and G*).

We then tested the egg retention and egg-laying behaviors of *Δdeedum* females compared to wild type to assess how the mutants compare to wild type in their reproductive resilience during drought. We blood fed wild type and *Δdeedum* mutant females, and withheld access to a freshwater substrate for either 6 days (*Figure 6E and F*) or 12 days (*Figure 6H, I*), corresponding to moderate or extended drought-like conditions. When we provided freshwater 6 days after the blood meal, 98% of wild type females compared to 90% of *Δdeedum* females laid at least one melanized egg (*Figure 6E and F*).

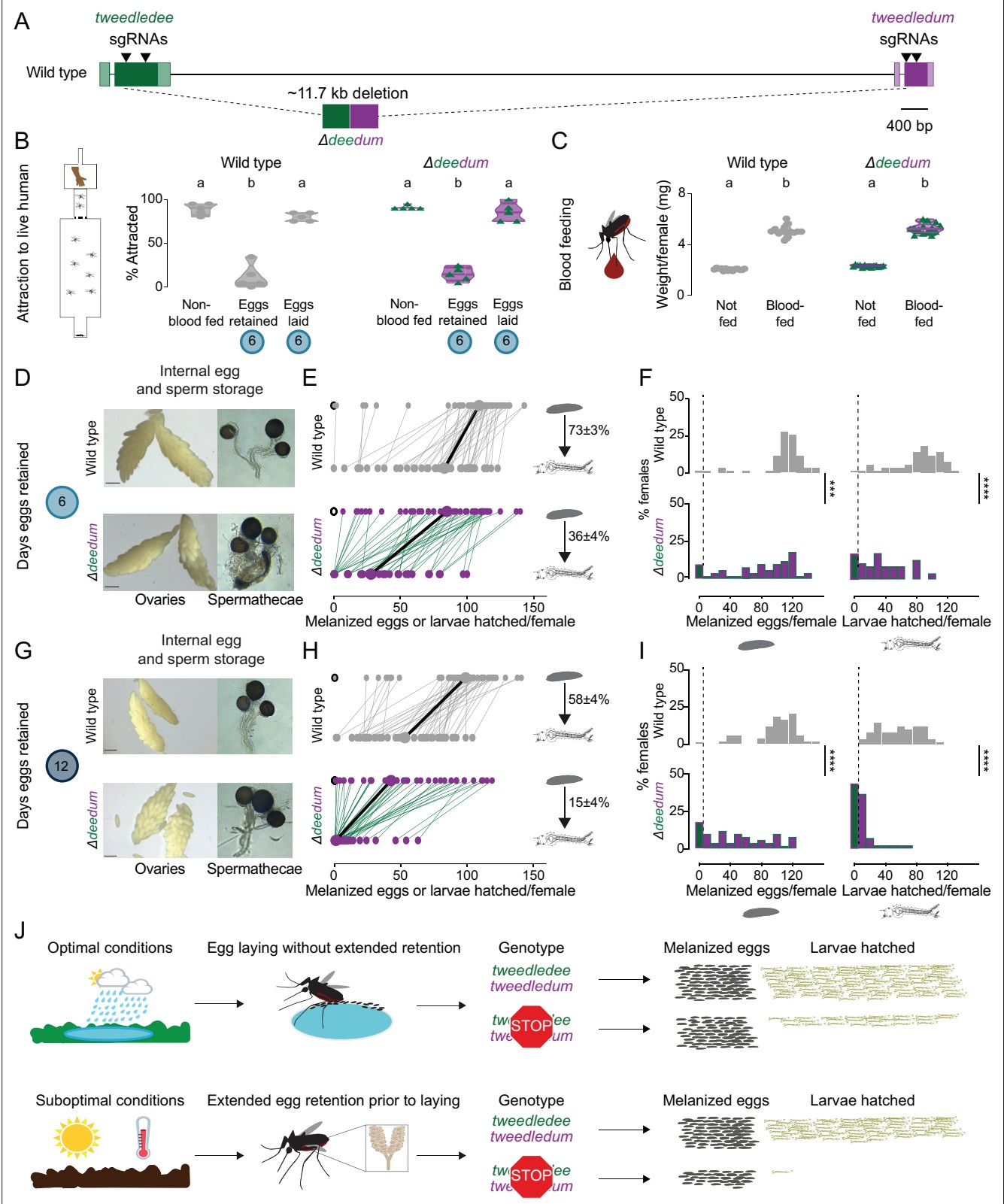

**Figure 6.** *tweedledee* and *tweedledum* are required for reproductive resilience during drought. (**A**) Schematic of *Δdeedum* mutant that deletes both *tweedledee* and *tweedledum*. (**B**) Attraction of wild type and *Δdeedum* mutant females to a human forearm. Data are plotted as violin plots with median and 1st/3rd quartiles and showing all data points. Each point represents a single trial with ~20 females, n=5 trials/group. Significantly different groups are indicated by different letters (one-way ANOVA, Holm-Šídák's multiple comparisons test, p<0.0001). (**C**) Averaged weights of 5 females of the

*Figure 6 continued on next page*

*Figure 6 continued*

indicated genotype not fed or blood fed, n=14 groups of 5 females per group. Data are plotted as violin plots with median and 1st/3rd quartiles and showing all data points. Significantly different groups are indicated by different letters (one-way ANOVA, Tukey's multiple comparisons test, p<0.0001). (**D, G**) Photographs of ovaries (left, scale bars: 20 µm) and spermathecae with filled sperm (right) from wild type and Δ*deedum* females 6 days (**D**) or 12 days (**G**) post blood-meal with eggs retained. (**E, H**) Number of melanized eggs laid by (top) and larvae hatched from (bottom) single wild type and Δ*deedum* mutant females 6 days (**E**) and 12 days (**H**) post blood-meal, depicting moderate and extended egg retention, respectively. Females laying no melanized eggs are depicted by open circles. Lines connect melanized eggs laid by and larvae hatched from the same individual. Larger circles and bold lines represent medians. Numbers at right show hatching rate (mean ± S.E.M) from each egg retention group, n=48–50 females/ group. (**F, I**) Distribution of melanized eggs laid (left) and larvae hatched (right) after egg retention in wild type and Δ*deedum* mutant females 6 days (**F**) or 12 days (**I**) post-blood-meal. 0 values are binned separately for each group. All other bins are [1,10], [11,20] … [101,110], [111,120] where "[,]" denote closed/ inclusive intervals. (**F, I**) The groups between each genotype for melanized eggs laid and larvae hatched respectively were compared at each of the time points to determine significant difference (Mann-Whitney tests, *** p<0.001; **** p<0.0001). Distributions in (**F**) are analyzed from data in (**E**) and distributions in (**I**) are analyzed from data in (**H**). (**J**) Summary of *tweedledee* and *tweedledum* function in drought resilience of female *Aedes aegypti* mosquitoes.

The online version of this article includes the following source data and figure supplement(s) for figure 6:

**Figure supplement 1.** Δ*deedum* double mutant genotyping strategy and additional egg-laying data.

**Figure supplement 1—source data 1.** Genotyping strategy for Δ*deedum* double mutants.

**Figure supplement 2.** *tweedledum* mutants show defects in reproductive resilience during drought.

While wild type females laid a median of 109 melanized eggs, Δ*deedum* mutant females laid a median of 85 melanized eggs. Of the females that laid any melanized eggs, 98% of the wild type females compared to 82% of Δ*deedum* females produced at least one viable offspring (*Figure 6E and F*). On average, the wild type hatch rate was 73%, while the hatch rate for Δ*deedum* eggs was only 36% after moderate egg retention (*Figure 6E and F*). Deleting *tweedledee* and *tweedledum* therefore had a considerable effect on egg viability during moderate egg retention.

If instead we withheld freshwater for 12 days after the blood meal before providing it to females, 98% of wild type females still laid at least one melanized egg compared to 82% of Δ*deedum* females (*Figure 6H, I*). Of the females that laid melanized eggs, 100% of the wild type females still produced at least one viable offspring compared to only 56% of Δ*deedum* females (*Figure 6H, I*). Wild type animals laid a median of 98 melanized eggs, with an average hatch rate of 58% (*Figure 6H, I*). Δ*deedum* females laid a median of 43 melanized eggs after this extended retention, but in stark contrast to wild type, they only had an average hatch rate of 15% (*Figure 6H, I*). Therefore, as the duration of drought increased, females lacking *tweedledee* and *tweedledum* very significantly lost their ability to remain reproductively resilient. At both 6 days (*Figure 6—figure supplement 1B*) and 12 days (*Figure 6— figure supplement 1C*) post-blood-meal, heterozygous females laid a similar number of eggs to wild type females, indicating that theΔ*deedum* phenotype is recessive.

*Aedes aegypti* females undergo profound changes in physiology and behavior upon mating (*Villarreal et al., 2018*; *Alfonso-Parra et al., 2016*; *Duvall et al., 2017*; *League et al., 2021*). To ask if the genotype of the male with which the female had mated had an influence on these female reproductive resilience phenotypes, we mated both wild type and Δ*deedum* females to wild type males instead of sibling males of their respective genotypes. We tested the number of melanized eggs laid by females after extended retention and found that Δ*deedum* females laid significantly fewer melanized eggs compared to wild type females (*Figure 6—figure supplement 1D*). These data show that the decreased fitness after egg retention seen in Δ*deedum* females is a maternally-derived phenotype.

In this study, we generated a deletion that disrupted both *tweedledee* and *tweedledum*. This left open the question of whether both genes contribute to reproductive resilience during drought. To date, we have been unable to establish a homozygous Δ*dee* single mutant strain. However, we recovered a Δ*dum* single mutant (*Figure 6—figure supplement 2A–B*) and observed that it had a phenotype largely overlapping with that of Δ*deedum* (*Figure 6—figure supplement 2C–I*). The Δ*dum* mutant had a deletion of 175 bp within the second exon, and the resulting gene fusion is predicted, in silico, to encode a protein with the first 17 amino acids of tweedledum conserved before the breakpoint junction, after which a frameshift adds 7 missense amino acids followed by a stop codon (*Figure 6—figure supplement 2A–B*). Δ*dum* mutants engorged on blood meals to approximately double their body weight (*Figure 6—figure supplement 2C*). They retained eggs and motile sperm in their visually healthy ovaries and spermathecae, respectively, for moderate (6 days post-blood-meal,

*Figure 6—figure supplement 2D*) or extended (12 days post-blood-meal, *Figure 6—figure supplement 2G*) durations post-blood-meal. *Δdum* single mutants laid a similar number of melanized eggs compared to wild type after moderate retention (*Figure 6—figure supplement 2E–F*), but significantly fewer melanized eggs after extended retention (*Figure 6—figure supplement 2H–I*). Of the melanized eggs laid after extended retention, a starkly smaller proportion from *Δdum* single mutants compared to wild type generated viable offspring (*Figure 6—figure supplement 2H and J*). These results suggest that at minimum *tweedledum* is contributing to drought resilience, and future work will resolve if both *tweedledee* and *tweedledum* are required for this important phenomenon.

## Discussion

### Reproductive flexibility enables a freshwater-centric lifestyle in variable environments

*Aedes aegypti* mosquitoes depend on freshwater availability for completing the aquatic larval and pupal stages of their life cycle (*Bentley and Day, 1989*; *Day, 2016*; *Wallis, 1954*). Adult *Aedes aegypti* females carrying mature eggs accordingly prefer to lay them at the edge of freshwater. Fluctuating climates with unpredictable and intense droughts likely impose selective pressures on this species, which has evolved multiple reproductive strategies that contribute to its resilience and invasive potential. Decoupling of mating from host-seeking and subsequent blood-feeding – such that either can take place first – and the appropriate coupling of both behaviors with egg-laying provides female *Aedes aegypti* mosquitoes with flexibility to maximize their reproductive output while still ensuring they have the required sperm and blood proteins for producing viable offspring. If faced with drought after being laid, embryos developmentally arrest within the eggshell for several months as an added layer of protection, until freshwater that can support larval survival becomes available to stimulate hatching (*Rezende et al., 2008*).

We suggest that the ability of adult *Aedes aegypti* females to retain mature eggs in their ovaries without complete loss of viability for flexible lengths of time while searching for an egg-laying site is of significant adaptive value. In this study, we demonstrated that *tweedledee* and *tweedledum*, a pair of linked, mosquito-specific genes, encode proteins that allow a female to retain her eggs for extended durations as needed without marked loss of viability, such as when access to freshwater is precluded due to drought. Females lacking both genes show a time-dependent phenotype. Their reproductive resilience dramatically worsens as length of egg retention increases from 6 to 12 days post-blood-meal, whereas wild type females continue to maintain a remarkable degree of reproductive resilience regardless of the duration of egg retention (*Figure 6J*). Our findings highlight an example of plasticity in the innate reproductive behaviors of *Aedes aegypti* mosquitoes, which allows them to thrive in a remarkable range of ecosystems with distinct climates.

### Providing protection to eggs: abundance in the right place, at the right time

Oocytes are stored within the ovaries of females in species separated by millions of years of evolution. Increased oocyte storage time carries the risk of increased damage, which females have evolved diverse strategies to mitigate (*Greenblatt et al., 2019*). Mammalian oocytes are maintained for decades and gradually released from their reserves. In humans, as a female and her oocyte reserve both age, oocytes become increasingly prone to meiotic segregation errors that result in higher rates of miscarriage and Down's syndrome (*Webster and Schuh, 2017*). Mammalian oocytes in reserve reside within primordial follicles where they are nurtured by maternally derived nutrients that play a role in maintaining oocyte longevity (*Webster and Schuh, 2017*). Bidirectional exchange between germline (oocyte) and somatic (follicle) cells is critical for germline maintenance and occurs through both gap junction-mediated transfer of small molecules, as well as via paracrine secretion of nourishing factors from follicles (*Kidder and Vanderhyden, 2010*). In *Drosophila melanogaster* flies, oocytes are retained if access to protein or sperm is restricted, and extended oocyte storage results in lower capacity for embryonic development. In wild type fly oocytes, abundant expression of two heat shock protein chaperones, *Hsp26* and *Hsp27* contributes to maintenance of developmental capacity following extended oocyte retention (*Greenblatt et al., 2019*). These examples highlight a strong

precedence for the existence of protective or nurturing mechanisms in *Aedes aegypti* ovaries, which would enable the female to maintain viable eggs for extended durations post-maturation.

What is the mechanism by which a pair of abundantly expressed genes ensure long-term viability of eggs retained in the mosquito ovary? We have few clues to work with. The highly regulated spatio-temporal expression of these paired genes in the ovary suggests to us that they both contribute to egg viability during retention. After maturation, and in the hours post-egg-laying, expression of *tweedledum* is restricted within the ovaries to the follicular epithelial cells surrounding eggs. In contrast, *tweedledee* is basally expressed in all the reproductive states of the ovaries, although its upregulation by several orders of magnitude is concurrent with its expanded expression in the follicular epithelial cells surrounding mature eggs during egg retention, together with *tweedledum*. Since the follicular epithelial cells form a socket around mature eggs, the expression of both genes together in these cells may be most functionally relevant in protecting eggs during retention, such as by forming a secreted desiccation-resistant barrier or a coating required for long-term maintenance. On the way to being laid, mature eggs also interact with the calyx where *tweedledee* is additionally expressed. It is therefore possible that additional interaction between the eggs and *tweedledee* while eggs are in transit facilitates egg competence for subsequent sperm entry and fertilization after extended retention, thereby ensuring viability. Future biochemical studies may reveal mechanistic insight into whether the proteins interact with each other, or with other molecules that provide protection to the egg after maturation.

In this study, we generated both Δ*deedum* double mutants and a Δ*dum* single mutant. Both strains have substantially similar phenotypes of reduced egg viability during extended retention. Our attempts to generate Δ*dee* single mutants were unsuccessful. This leaves open the question of whether both *tweedledee* and *tweedledum* contribute to the Δ*deedum* phenotype or whether *tweedledum* is solely responsible.

## Signal, waste, or both?

What is the function of the circulating forms of these proteins? Our proteomic analysis supports the hypothesis that these proteins are secreted, as we detect tryptic peptides from both proteins that represent N-terminal signal peptide-cleaved forms, both in the circulating hemolymph, as well as in the ovary. It is unclear if the circulating forms of these proteins are waste products destined for destruction after production in the ovaries, or if they serve a signaling function. One possibility is that tweedledee and tweedledum sustain eggs in the ovary while also acting as a humoral signal to alert other organs – including the nervous system – of the female's reproductive status. There is a precedent for secreted egg-related proteins serving multiple roles. Vitellogenins are best known as egg yolk proteins, but also function as a hormone secreted into hemolymph, where they have multiple effects on behavior and longevity in social insects (*Corona et al., 2007*). In *Aedes albopictus*, vitellogenins have recently been implicated in regulating host-seeking behavior in response to the status of nutritional reserves, in addition to their role as yolk protein precursors (*Dittmer et al., 2019*). Bifunctionality and coordinated action of molecular pairs are common phenomena in insect reproduction. The hormones, 20-hydroxyecdysone and juvenile hormone III, act in concert to control metamorphosis across insects (*Jia et al., 2017*), while additionally modulating vitellogenesis after a blood meal in mosquitoes (*Hansen et al., 2014*; *Raikhel and Lea, 1991*; *Roy et al., 2015*) and defining caste-specific behavioral repertoires or traits in the ant, *Harpegnathos saltator* (*Gospocic et al., 2021*).

## Taxon-restricted genes underlie adaptations in a diverse world

Adaptations, especially those relevant to reproduction or expansion into new ecological niches, have been shown in recent studies across diverse species to arise from taxon-restricted, rapidly evolving genes with tissue-restricted and/or sexually dimorphic expression (*Schmidt et al., 2013*; *Guillén et al., 2014*; *Witt et al., 2021*). For example, a mouse de novo gene, *Gm13030*, with female-biased, oviduct-specific expression shows strong estrous cycle-dependent control (*Xie et al., 2019*). In homozygous *Gm13030* mutant females, three *Dcpp* genes known to promote embryo implantation are upregulated. Mutant females progress normally through their first estrous cycle but undergo premature implantation in their second estrous cycle, resulting in inappropriately early second litters and higher infanticide rates – both likely maladaptive phenotypes (*Xie et al., 2019*). In *Rhagovelia* water strider insects, a pair of taxon-restricted genes (*gsha* and *mogsha*) are required for the development

of a midleg fan structure specific to this genus (*Santos et al., 2017*). The fan endows *Rhagovelia* with the biomechanical capabilities needed to perform a rowing behavior on the surface of rapidly moving streams where they are typically found (*Santos et al., 2017*). Other non-*Rhagovelia* species occupying the same streams rarely perform these rowing behaviors and instead occupy static surfaces on leaves, suggesting the fan is central to *Rhagovelia's* ability to walk on water (*Santos et al., 2017*). *Hormaphis cornu* aphids secrete salivary gland-enriched BICYCLE proteins, which come from a large family of rapidly evolving secreted molecules. The aphids pierce their stylet into mesophyll cells of the witch hazel leaf, where they deposit BICYCLE proteins. This triggers the formation of galls on the witch hazel leaf, which provides the aphid with the shelter and nutrition required for subsequent development (*Korgaonkar et al., 2021*). *Aedes aegypti* and *Aedes albopictus* mosquitoes are both predicted to expand into new parts of the globe where they were previously absent, owing to their ecological flexibility (*Ryan et al., 2019*). Our case study of two taxon-restricted genes, *tweedledee* and *tweedledum,* adds support that some rapidly evolving genes may be functionally important to allow the ecological flexibility of species without compromising reproductive capacity. Although the strong correlations between the variations of the two genes and ecological variables are not direct evidence showing that the two genes are adaptive, it indirectly supports the role of these genes in climate-related variables. Since we used a candidate-gene approach to study the functions of these taxon-restricted genes, we do not reject the possibility that the divergence of many conserved genes, as shown in myriad studies, may be functionally important to adaptation (*Colosimo et al., 2005*; *Manceau et al., 2011*).

## Evolutionary origins of *tweedledee* and *tweedledum*

The evolutionary history of *tweedledee*, *tweedledum*, and the *putative orthologs* is intriguing. The lack of known domains poses a problem for understanding the function and evolutionary history of these proteins, and conventional homology searches fail to detect homologs that could provide clues. The two proteins have no homology to each other, to the putative orthologs, or to any other known protein. Structural homology-based approaches might be a path forward to identify genes with divergent sequence but conserved three-dimensional protein structure and function. However, current protein structure prediction programs perform poorly on small proteins, especially when phylogenetic homology cannot guide the analysis (*Jumper et al., 2021*).

Rapidly evolving genes typically show testis-biased expression across evolution (*Witt et al., 2021*), but in *Aedes aegypti,* ovary-specific genes evolve unusually fast with more frequently occurring signatures of positive selection as compared to genes with enriched expression in the testis (*Whittle and Extavour, 2017*). Our study reveals *tweedledee* and *tweedledum* as examples of such rapidly evolving, ovary-enriched genes present in mosquito genomes, but it remains an open question whether genes of this type exist outside of mosquitoes. Published transcriptomic data suggest the tissue-restricted and sexually dimorphic expression of these genes may be conserved across genera. In *Anopheles stephensi* (*Jiang et al., 2014*), the *putative ortholog* is upregulated in ovaries 24 hr post-blood-meal, and in *Anopheles arabiensis,* the *putative ortholog* is upregulated in the reproductive tissues of females compared to males (*Papa et al., 2017*). At the genomic level, what allows this syntenic locus, characterized in several blood-feeding mosquitoes by the conserved *scratch* and *peritrophin-like* genes, to 'trap' one rapidly divergent gene in the case of *Culex* and *Anopheles* mosquitoes, or two rapidly divergent genes in the case of *Aedes* mosquitoes? Together, these observations of shared synteny and gene expression indicate that the *Aedes tweedledee/tweedledum,* and the *Culex* and *Anopheles putative orthologs* are likely to have evolved from a common ancestor, and that these genes may co-opt similar pathways to function across genera.

A look at the natural histories of different mosquito genera suggests that they each employ distinct life history strategies. These involve differences in adult female behavioral parameters: flexibility in choosing a blood meal host, circadian control of host-seeking, mating frequency, and egg-laying site selection; differences in the potential for diapause and dormancy/quiescence or resistance to desiccation in embryos; and tolerance of larvae for different aquatic environments (*Degner and Harrington, 2016*; *Wallis, 1954*). The physiological and behavioral adaptations underpinning these reproductive strategies must co-evolve in each of the species, in turn determining the ecological niches that the species are able to exploit. Future comparative studies will resolve whether rapid divergence in the sequence of *tweedledee/tweedledum* and the *putative ortholog* is accompanied by conservation, or

by rapid divergence in their functions. This work thus highlights the importance of considering taxon-restricted genes as important points of study to understand the life-history strategies of a species, and to identify new inroads for breaking the cycle of mosquito-borne disease transmission.

Although many factors may affect the power for detecting correlations between allele frequencies among populations and ecological variables (*Coop et al., 2010*), both *tweedledee* and *tweedledum* are strongly correlated with a few important variables related to temperature and/or precipitation. Furthermore, the permutation tests support the hypothesis that the polymorphisms of the two genes in the populations are more impacted by their environments compared to other genes encoded by the *Aedes aegypti* genome, suggesting that some of the mutations are adaptive and thus under selection. In future work, analyses of additional *Aedes aegypti* populations collected from regions with disparate climates, coupled with comparative studies of egg retention capacities, could provide exciting insights into the functional relevance of different genetic changes and the selective pressures driving rapid evolution at this locus.

## Materials and methods

### Mosquito rearing and maintenance

*Aedes aegypti* wild type (Liverpool) and CRISPR-Cas9 knockout strains were reared using standard insectary conditions in an environmental chamber maintained at 70–80% relative humidity and 25–28°C with a photoperiod of 14 hr light: 10 hr dark as previously described (*DeGennaro et al., 2013*). Adults of all genotypes were provided ad libitum access to 10% sucrose and were housed in 30 cm³ BugDorm-1 Insect Rearing Cages (MegaView Science) unless otherwise specified. Newly generated mutant strains were blood-fed on human volunteers until they were established. For stock maintenance, females were blood-fed on live mice or on defibrinated sheep blood (Hemostat Laboratories, DSB100) using an artificial membrane feeder (the 'blood puck') described below. All animals used for behavior experiments, regardless of genotype, were blood-fed using the blood puck.

### Blood-feeding for behavior assays using the blood puck

For all behavior experiments requiring blood-fed mosquitoes, 5- to 16-day-old females were fed defibrinated sheep blood supplemented with 2 mM adenosine 5'-triphosphate (ATP) (Sigma Aldrich, A6419) in aqueous sodium bicarbonate buffer using a new artificial membrane feeder we designed called the blood puck. Metal blood pucks were custom-made at The Rockefeller University Precision Instrumentation Technologies Resource Center and the Rockefeller High-Energy Physics Machine Shop. Three-dimensional designs for fabrication, and a bench manual for suggested use are provided (https://doi.org/10.5281/zenodo.5945524). The blood puck is a disc with one indented, rimmed face on which blood rests with Parafilm stretched over it. This allows the female mosquitoes to pierce the Parafilm membrane and feed on the blood beneath. The other face of the disc is fully flat and does not have Parafilm stretched across its surface.

Before assembling the blood puck, 8.1 mL of defibrinated sheep blood stored at 4 °C was warmed to 42 °C for 15–30 minutes in a water bath, and 1 mL aliquots of 20 mM ATP in 25 mM aqueous sodium bicarbonate stock stored at –20 °C were slowly thawed on wet ice to room temperature. To assemble the feeding disc of the blood puck membrane-feeder, a 10x10 cm square of Parafilm M (Fisher Scientific, S37440) was first rubbed on both sides against a human skin surface free of cosmetics, such as the forearm or neck, then stretched evenly until translucent before setting aside. The blood puck disc was placed in a metal bead or water bath at 42 °C for at least 10 min. It was then removed from the warming bath and thoroughly dried with a paper towel. Next, the Parafilm rubbed on human skin was stretched across the entire indented face of the disc with the utmost care taken to ensure there were no holes in the Parafilm on the feeding side of the disc. Additional strips of Parafilm were used to seal the edges of the disc, and the stretched Parafilm was checked to ensure that it was taut enough to be pierced by a female mosquito's stylet. Working quickly to prevent heat dissipation from the pre-heated feeding disc and blood, 900 µL of ATP stock was added to the 8.1 mL of heated blood for a final concentration of 2 mM ATP, and vortexed thoroughly to mix. Care was taken to ensure the ATP was never heated and did not undergo multiple freeze-thaw cycles. The blood puck disc was held by its edges with the indented, rimmed side face-down and the flat side face-up. The blood + ATP mixture was pipetted through one of the two holes from the flat face. The disc was swirled laterally

to evenly distribute the blood before gently placing the blood puck on top of a mesh face of the mosquito cage. In this configuration, the indented, rimmed side sat atop mesh of the cage with female mosquitoes beneath, while the flat side was face-up. Any excess blood dribbling out of the puck after placing on the mosquito cage was blotted with paper towels, and 1–2 additional metal discs (either additional blood pucks, or simple metal discs with both faces flat) pre-warmed to 42 °C were placed on top of the feeding disc to maintain warmth. These discs were reheated and replaced as needed to maintain the feeding disc at an optimal temperature for mosquito blood-feeding. As needed, mosquitoes were activated by an experimenter exhaling their breath into the cage. Blood-feeding was conducted both at ambient room temperature conditions and in the environmental chamber with similarly high and reliable engorgement rates. Females were typically allowed to feed for 15 min, or until fully engorged, and a single blood puck could be used for 2–3 cages of ~400–450 females each with replacement of rewarmed flat discs between transfer of the apparatus between cages. After feeding to repletion, typically within 15–30 min per cage, the discs were taken off the cage, the Parafilm discarded into biohazard waste, and the metal discs rinsed under hot water to thoroughly remove all traces of blood. The blood puck was dried with paper towels for subsequent use.

After feeding, animals were cold anesthetized in a 4 °C cold room to separate and discard males, as well as non-blood-fed and partially engorged females. Fully engorged females were selected by eye and returned to their original rearing conditions in a fresh cage with continuous access to 10% sucrose.

## Preparation of mosquitoes for weighing

When blood meal size was measured by weighing (*Figure 6C*, *Figure 6—figure supplement 2C*), non-blood-fed females of all genotypes were each split into two cages of 80–100 females and sugar-starved for 20–24 hr prior to delivering the blood meal. During the sugar-starvation period in experiments involving subsequent weighing, females were offered deionized water-soaked cotton balls to prevent dehydration. Non-blood-fed controls and experimental group females engorged on blood were both immediately cold anesthetized at 4 °C after offering the blood meal and weighed in respective groups of 5 each, as previously described (*Jové et al., 2020b*).

## Preparation of mosquitoes at different reproductive time-points

To prepare groups of mosquito females at different reproductive time-points, all groups were age-matched within each experiment to the extent possible and maintained in mixed-sex cages for at least 5–7 days post-eclosion to ensure that most females were mated. The only exception was with the "virgin" group (*Figure 1B and F*), for which females were separated at the pupal stage and maintained in single-sex cages. For all blood-feeding groups, females were provided sheep blood supplemented with 2 mM ATP, and only fully engorged females were selected by eye for subsequent experimental use. For egg retention groups, cages were carefully checked for any prematurely dumped eggs prior to collection of females for dissections. For all experiments in which females were required as soon after egg-laying as possible, that is, groups where eggs were laid <5 hr prior (*Figure 1—figure supplement 1C–D*, *Figure 2*, and *Figure 3*), we standardized 3 hr as the allotted time for individual females after they were aspirated into egg-laying vials at room temperature. The allotted time of 3 hr was determined based on our finding that ~80% of females complete egg-laying within 3 hr of transfer to egg-laying vials (*Figure 1—figure supplement 1A–B*). Eggs laid by *Aedes aegypti* females are initially white, and melanize within the first 1–2 hr of egg-laying (*Isoe et al., 2019*). Based on this, we postulated that any egg-laying vials with at least 10 melanized eggs are likely to have come from females that had completed laying their full clutch of ~100 eggs.

Females of the <5 hr post-egg-laying group used for ovary RNA-seq, ovary proteomics, and hemolymph proteomics experiments in *Figure 2* were tested behaviorally to verify that they had restored attraction to humans using a long-range, live human stimulus olfactometer as described (*Basrur et al., 2020*). Briefly, females that had laid ≥10 melanized eggs were pooled into reproductively similar groups of 20, gently aspirated at room temperature into 'start' canisters of the olfactometer, acclimated for 10 min, and the trial run for 5 min and 30 s as per standard assay protocol. Attracted females were defined as those that entered the trap proximal to the human arm. These attracted females were collected directly from the trap, and only these attracted females were dissected for

ovary or hemolymph sample collection. Females that did not enter the attraction trap were discarded and not used for ovary or hemolymph sample collection.

For whole-mount ovary fluorescence RNA in situ hybridization experiments in *Figure 3B–E*, females were collected and used immediately from egg-laying vials that contained at least 10 melanized eggs, without further assessment of their attraction to humans. For groups that had laid eggs greater than 1 week prior to sample collection (*Figure 3F*), plastic cups (VWR HDPE Multipurpose Containers, H9009-664) half-filled with deionized water and lined with filter paper (GE Healthcare, WHA1001055) were introduced to the cage as continuously available egg-laying substrates between 3 and 6 days after blood-feeding. When dissected, 13 days had elapsed since the last blood-meal of this group. When groups in their second reproductive cycle were collected, either for behavior (*Figure 1E*) or for hemolymph proteomics (*Figure 2K–M*), they were treated equivalently to the corresponding first reproductive cycle groups.

## Live human olfactometer assay

Live human olfactometer assays for testing female mosquito attraction to a human forearm were performed as previously described (*Basrur et al., 2020*). The same subject was used as a stimulus in all experiments. Fabrication and assembly details, as well as a user guide are available at https://github.com/VosshallLab/Basrur_Vosshall2020 (*Vosshall Lab, 2020*). Experiments were conducted at 70–90% humidity and 25–28°C. Each trial consisted of approximately 20 female mosquitoes, grouped by reproductive condition. The groups of 20 were aspirated into 'start' canisters at least 30 min before their trial. Females were given continuous access to 10% sucrose prior to sorting into canisters but were not provided any sucrose or water after being re-housed in the canister. Trials with non-blood-fed females were treated as positive controls and were interspersed throughout each experimental day. Experimental days were counted for final analysis only after ensuring average attraction to the live human arm stimulus of the non-blood-fed group was ≥50% across trials. All groups were run on each experimental day. Two trials were run simultaneously, one using each arm of the live experimenter as the stimulus. Groups were shuffled between ports to minimize bias.

## Egg retention, laying, and hatching assay

For all egg retention experiments, to prevent accumulated condensation that could trigger premature egg 'dumping', special care was taken to ensure that all cages and sucrose-soaked cotton wicks for blood-fed female mosquitoes were not subjected to frequent fluctuating temperature and humidity, and closely monitored to remove any accumulated droplets of water. Following the duration of egg retention, cages were thoroughly checked for any dumped eggs. If a small proportion of eggs was found prematurely dumped on the sugar-soaked cotton wicks, this was noted prior to set up of egg-laying. Although observed extremely rarely, if the cage floor was found to be covered with a large proportion of prematurely dumped eggs, suggesting that little to no egg retention was achieved, all females in such a cage were discarded and not used for further experimentation.

Only for the data shown in *Figure 6—figure supplement 1D*, a single-female modular egg-laying assay setup was used as described (*Matthews et al., 2019*). The assay setup was modified to accommodate 28 females instead of 14, and each female was provided access to a single egg-laying substrate of deionized water instead of two substrate choices. Details of design and fabrication are available at https://github.com/VosshallLab/MatthewsYoungerVosshall2018 (*Vosshall Lab, 2019*).

For all other egg-laying behavior experiments, at the time of egg-laying, females with retained eggs were 14–21 days old, and aspirated out of their cages at room temperature into individual egg-laying vials. Egg-laying vials were made using plastic *Drosophila* vials (VWR, 25 mm diameter, 95 mm length, 75813–164) with 2–3 mL of deionized water, and a 55 mm diameter Whatman filter paper (GE Healthcare, WHA1001055) folded into a cone at the bottom of the vial to serve as a moist egg-laying substrate as previously described (*Matthews et al., 2019*; *Duvall et al., 2019*). Vials were kept plugged (Genesee Scientific, Flugs Narrow Plastic Vials, 49–102) under standard insectary conditions following transfer of females ready for egg-laying. All females were removed 20–24 hr after transfer under brief cold anesthesia and either discarded or stored at –20 °C if required for further genotyping. Filter paper lined with laid eggs, and any eggs remaining on the sides of the vial or in the water, were removed from the vial at room temperature and placed briefly on a paper towel to remove excess moisture. The melanized eggs were then manually counted by eye or under a dissection scope as

needed and the number of melanized eggs laid per female recorded. If most eggs from a female were unmelanized or submerged in the pool of water instead of lined on the filter paper, the sample was excluded. The egg-lined filter paper was returned to the emptied and dried vial within 24 hr of removing the female and terminating the egg-laying assay. All vials were kept under standard insectary conditions for 6–14 days prior to hatching.

Egg hatching was staggered to ensure that all egg papers were dried for the same length of time prior to hatching (6–14 days), and egg papers from distinct individuals were hatched and maintained separately. Eggs were hatched either by transferring egg papers into a small plastic cup (VWR, HDPE Multipurpose Containers, H9009-662) with 50–60 mL 'hatch broth' comprised of deoxygenated water with finely ground fish food (Pet Mountain, Tetramin Tropical Tablets Fish Food for Bottom Feeders, YT16110M), or by adding 20 mL of hatch broth directly to the egg-laying vial with the dry egg paper (*Figure 1*, and *Figure 6* or *Figure 6—figure supplement 2*, respectively). Larvae hatched were provided with a fresh pinch of fish food as needed.

At least 5 days after hatching, the egg viability experiment was terminated. Egg papers were removed and larvae, sometimes mixed with pupae or eclosed adults, were either cold anesthetized at 4 °C or killed by freezing at –20 °C overnight before thawing and counting. Offspring from each individual female were separately poured onto Petri plates (Fisher Scientific, S33580A) or small plastic cups with fresh deionized water and photographed on a light board using a webcam (Logitech, C922x Pro Stream Webcam) mounted from above, ensuring that all offspring were captured in the field-of-view. Captured images were imported into FIJI/ImageJ (NIH), and the number of offspring from each individual female was counted using the Cell Counter plugin.

## Bulk RNA-sequencing of mosquito ovaries

For ovary bulk RNA-sequencing (RNA-seq), 3 pairs of ovaries were used for each replicate, and 4 replicates were prepared per experimental group from 19-to-20-day old females. Mosquitoes were cold-anesthetized and kept on ice for up to 1 hr, or until dissections were complete. Ovaries were dissected on ice, in ice-cold RNase-free 1 X phosphate-buffered saline (PBS) (Invitrogen, AM9625). They were moved using forceps into 0.5 mL Eppendorf LoBind microcentrifuge tubes (Sigma Aldrich, Z666521), and immediately snap-frozen on a cold block (Simport, S700-14) pre-chilled to –78 °C on dry ice. Extreme caution was taken during the tissue dissection to ensure that there was no contamination from other mosquito tissues. Each dish and forcep was carefully cleaned with 70% ethanol and RNase-away (Thermo Fisher, 7003) after every dissection. All replicates for each experimental group were dissected in parallel to avoid artifacts and batch effects. Dissected tissue was stored at –80 °C until RNA extraction.

RNA extraction was performed using the PicoPure Kit (Thermo Fisher, KIT0204) with the following modification for homogenizing tissue: instead of lysis buffer, 100 µL of TRIzol (Thermo Fisher, 15596026) was added to the collection tube on ice. Tissues were homogenized manually using a Pellet Pestle Motor (Kimble, 749540) and an RNase-Free pellet pestle (VWR, KT749510-0590) for 30 s following the addition of 140 µL of TRIzol to a total of 240 µL. Tubes stood at room temperature for 5 min before 48 µL of chloroform:isoamyl alcohol 24:1 was added (Sigma, C0549). Tubes were hand-shaken for 30 s and left to stand for 2 min before centrifuging at 12,000 RPM for 15 min at 4 °C. The aqueous TRIzol layer was then removed and added into the PicoPure column, up to 130 µL at one time. Subsequent steps were performed according to PicoPure manufacturer's instructions, including DNase treatment.

Samples were run on a Bioanalyzer RNA Pico Chip (Agilent, 5067–1513) to determine RNA quantity and quality. RNA quantity was re-verified with a Qubit 2.0 Fluorometer using the RNA HS Assay Kit (Invitrogen, Q32855). The three biological replicates with the most consistent RNA yield across conditions were then used for library preparation and sequencing.

Of total RNA, 100 ng was used to generate RNA-seq libraries using the Illumina TruSeq stranded mRNA LT kit (Illumina, 20020594), following the manufacturer's protocol. Libraries prepared with unique dual indexes were pooled at equal molar ratios. Sequencing was performed at The Rockefeller University Genomics Resource Center on the Illumina NovaSeq 6000 sequencer using V1.5 reagents, the SP flow cell, and NovaSeq Control Software V1.7.0 to generate 150 bp paired end reads, following manufacturer protocol. Data were demultiplexed and delivered as fastq files for each library. Sequencing reads have been deposited at the National Center for Biotechnology Information (NCBI) Sequence Read Archive (SRA) under BioProject PRJNA796320. The data discussed in this

publication have been deposited in NCBI's Gene Expression Omnibus (*Edgar et al., 2002*) and are accessible through GEO series accession number GSE193470.

## Alignment and quantification of RNA-seq data

Sequence and transcript coordinates for the *Aedes aegypti* mosquito genome and gene models were obtained by merging the Aaeg_L5 RefSeq annotation from NCBI with a manual chemoreceptor annotation. Information related to generating this annotation is available at https://github.com/VosshallLab/Jove_Vosshall_2020/tree/master/RNAseq_merged_annotation (*Jové et al., 2020a*). Transcript expression was calculated using the Salmon quantification software (version 0.8.2) (*Patro et al., 2017*), and gene expression levels as transcripts per million (TPMs) and counts were retrieved using Tximport (version 1.8.0) (*Love et al., 2016*; *Love et al., 2018*). A table of TPM counts for all reproductive conditions and replicates can be found on Zenodo (https://doi.org/10.5281/zenodo.5945524). Normalization and rlog transformation of raw read counts in genes were performed using DESeq2 (version 1.20.0) (*Love et al., 2014*). The normalized and transformed counts were used to perform principal component analysis (PCA) using DESeq2, and to assess between-sample variability with hierarchical clustering and with calculation of sample distance correlations (*Love et al., 2014*).

## Ovary collection and sample preparation for proteomics

To extract whole proteins from ovaries, 8 pairs of ovaries were used for each replicate, and 4 replicates were prepared per experimental group. Ovaries were dissected in a droplet of 1 X PBS, as needed, and boiled for 5 min at 100 °C in 150 µL MilliQ water. Samples were centrifuged at 12,000 RPM for 30 s. The water fraction was then decanted into a separate tube and set aside. Extraction solution (150 µL of 0.25% acetic acid) was added to the precipitate, and the tissue was homogenized with a 5 mm tungsten carbide bead in a bead mill homogenizer (Qiagen, Tissue Lyser II) at 30 Hz for 1.5 min. The water and acid fractions were centrifuged separately at 4 °C, 8000 RPM for 30 min. The two supernatants were then combined and spun to dryness in an Eppendorf Speedvac at 60 °C for 1–1.5 hr. The mass spectrometry proteomics data have been deposited to the ProteomeXchange Consortium via the PRIDE (*Perez-Riverol et al., 2019*) partner repository with the dataset identifier PXD030925. Ovary sample raw files begin with the code "MS205850LUM".

## Hemolymph collection and sample preparation for proteomics

To collect hemolymph, 5 females were used per replicate. Cold anesthetized females were kept on ice and decapitated using 2.5 mm cutting edge Vannas spring scissors (Fine Science Tools, 15000–08) under a dissection microscope at 10 X. Cold 30 µL1X PBS with 0.05% Tween (PBS-T) was pipetted as a bubble onto a 35 mm Petri plate (Falcon, 351008) on ice. The decapitated thorax was positioned close to the droplet without touching, and the thorax was gently squeezed using blunt forceps to release hemolymph from the decapitation site into the droplet of PBS-T. This was repeated such that each droplet of PBS-T consisted of pooled hemolymph from a total of 5 females for a single replicate. The PBS-T with hemolymph was pipetted into a 1.5 mL Eppendorf Protein LoBind tube (Thermo Fisher), and the Petri dish was washed with 10 µL PBS-T, and the 10 µL wash was combined with the ~30 µL from the initial extract. The samples were then heat-inactivated at 90 °C for 10 min, snap-frozen on dry ice and stored in Eppendorf Protein LoBind tubes at –80 °C until the subsequent steps could be carried out. For acetone precipitation of the extracted proteins, we ensured all samples, reagents, tubes, and tube racks were maintained at –20 °C. Hemolymph samples from –80 °C were quickly removed onto racks cooled to –20 °C after which 6 volumes of acetone (~210 µL) cooled to –20 °C were added. The sample tubes were vortexed for a few seconds until the frozen hemolymph samples fragmented and mixed well with the acetone. The tubes were then incubated upright at –20 °C overnight. Following incubation, samples were spun down at 13,000 x g at 4 °C for 10 min in a tabletop microcentrifuge. Most of the supernatant was removed with a pipette and discarded, leaving the protein pellet wet before storing at –80 °C until subsequent steps. The mass spectrometry proteomics data have been deposited to the ProteomeXchange Consortium via the PRIDE (*Perez-Riverol et al., 2019*) partner repository with the dataset identifier PXD030925. Hemolymph sample raw files begin with the code 'MS195106LUM'.

## Liquid chromatography-mass spectrometry (LC-MS)

Dry protein pellets of both ovary and hemolymph samples were dissolved and reduced in 8 M urea (Fisher Scientific, 45000234)/70 mM ammonium bicarbonate (Fisher Scientific, 501656826)/20 mM dithiothreitol (Sigma Aldrich, 233153), followed by alkylation in the dark with 50 mM iodoacetamide (Sigma Aldrich, I1149). Samples were then diluted twofold and digested overnight with endoproteinase LysC (Fujifilm Wako Chemicals, WAKA Lysyl Endopeptidase, 129–02541). Samples were additionally diluted twofold and digested with trypsin (Promega, Sequencing Grade Modified Trypsin, Lyophil, PRV5111) for 6 hr. Digestions were halted by acidification and peptides were solid phase-extracted prior to analysis by LC-MS/MS. Peptide samples were analyzed by nano-flow LC-MS/MS (EasyLC 1200) coupled to a Fusion Lumos (Thermo Fisher) operated in High/High Data Dependent Acquisition (DDA) mode using Lock mass m/z 445.12003. Peptides were separated by reversed phase chromatography using 12 cm/75 µm, 3 µm C 18 beads (Nikkyo Technologies, NTCC-360/75-3-123 Column) with buffer A: 0.1% formic acid (Fisher Scientific, A11750), and buffer B: 80% acetonitrile (Fisher Scientific, A955) in 0.1% formic acid. For the hemolymph samples, a gradient from 2% buffer B/98% buffer A to 35% buffer B/65% buffer A in 70 min was used. For the ovary samples, a gradient from 2% buffer B/98% buffer A to 38% buffer B/62% buffer A in 90 min was used.

Data were queried against 'GCF_002204515.2_AaegL5.0_protein.fasta' database using MaxQuant software with the Andromeda search engine v.1.6. 6.0 (*Cox et al., 2014*). Oxidation of methionine and N-terminal protein acetylation were allowed as variables, and cysteine carbamidomethylation was defined as a fixed modification. Mass tolerance was set at 4.5 parts per million (ppm) for precursor ions and 20 ppm for fragment ions. Two missed cleavages were allowed for specific tryptic database searches. The 'match between runs' setting was enabled. False discovery rate (FDR) for proteins was set at 1% combined with a peptide FDR of 2%. Intensity based absolute quantitation (iBAQ) (*Schwanhäusser et al., 2011*) values were used as a proxy for protein abundances. Data were processed using Perseus v.1.6.10.50 (*Tyanova et al., 2016*). Reverse database hits and contaminating proteins were removed, and it was required that a protein was to be measured (using iBAQ) in at least 3 of 4 replicates for at least one of the experimental groups. Each $\log_2$-transformed iBAQ signal was normalized by subtracting the respective sample's median iBAQ signal. Missing values were assumed 'Missing Not At Random' (MNAR) (*Lazar et al., 2016*) and a random distribution of signals with a width of 0.3 and a downshift of 1.8 were used to impute missing values. The sample sets were assessed for quality and correlation using scatter plots and PCA. Tables of iBAQ values and other analyzed metrics are available on Zenodo (https://doi.org/10.5281/zenodo.5945524) for all reproductive conditions and replicates for both ovary and hemolymph proteomics datasets.

## Whole-mount ovary fluorescence RNA in situ hybridization

The previously described hybridization chain reaction (HCR) technique (*Choi et al., 2018*; *Herre et al., 2022*) was modified to detect RNA in whole-mount ovaries. All reagents, including custom probes, amplifiers, Probe Hybridization Buffer, Amplification Buffer, and Probe Wash Buffer, were purchased from Molecular Instruments. Adult female mosquitoes were dissected ~20 days post-eclosion. They were grouped by reproductive condition, cold anesthetized at 4 °C, and maintained on ice for 30 min while ovaries were dissected from each female in 0.1 X PBS. Dissected ovaries were incubated in a solution of 4% paraformaldehyde, 1 X PBS, and 0.03% Triton X-100**,** and rotated overnight at 4 °C. Ovaries were then washed four times in 1 X PBS containing 0.1% Tween-20 (0.1% PBS-T) for 10 min each. Subsequently, ovary samples were dehydrated on ice using a series of graded methanol/0.1% PBS-T washes for 10 min, as follows: 25% methanol in 0.1% PBS-T, 50% methanol in 0.1% PBS-T, 75% methanol in 0.1% PBS-T, and two washes in 100% methanol. Ovaries remained in 100% methanol at –20 °C overnight. To rehydrate the ovaries, samples were washed for 10 min each on wet ice with a series of graded methanol/0.1% PBS-T solutions, as follows: 75% methanol in 0.1% PBS-T, 50% methanol in 0.1% PBS-T, 25% methanol in 0.1% PBS-T, and two washes of 0.1% PBS-T. Ovary tissue was then digested in 20 µg/mL Proteinase-K (Thermo Fisher, AM2548) with 0.1% PBS-T for 30 min at room temperature and subsequently washed twice in 0.1% PBS-T at room temperature for 10 min each. Tissues were then fixed in 4% paraformaldehyde in 0.1% PBS-T for 20 min at room temperature and washed three times in 0.1% PBS-T for 15 min each at room temperature.

Ovaries were incubated in Probe Hybridization Buffer for 5 min at room temperature, and subsequently in a 37 °C hybridization oven for 30 min in pre-warmed Probe Hybridization Buffer. A solution

of pre-warmed Probe Hybridization Buffer and probe sets, each at 8 µmol, was mixed, and used to incubate samples at 37 °C in a hybridization oven for three nights. Ovaries were next washed five times for 10 min each in a 37 °C hybridization oven using Probe Wash Buffer pre-warmed to 37 °C. The samples were then washed twice with 5 X saline-sodium citrate (SSC) buffer (Invitrogen, 15557044) containing 0.1% Tween-20 solution for 5 min each at room temperature. To pre-amplify, ovaries were incubated in room temperature Amplification Buffer for 10 min. A total of 24 µmol hairpins were prepared by heating 8 µL of 3 µM stock of H1 and H2 hairpins, separately, each at 95 °C for 90 s on an Eppendorf Mastercycler. The hairpins were cooled to room temperature for 30 min in the dark, as hairpins are photosensitive and subject to photobleaching. Hairpins were then added to 100 µL of Amplification Buffer in which ovaries were incubated on a rotator at room temperature in the dark overnight. Ovaries were next incubated in the dark in a solution of 1:1000 DAPI in 5 X SSC with 0.1% Tween-20 at room temperature for 1 hr. Ovaries were finally washed four times for 10 min each in 5 X SSC with 0.1% Tween-20 and mounted in SlowFade Diamond (Thermo Fisher, S36972) onto glass slides with confocal microscopy-compatible coverslips.

### Ovary confocal imaging

Images were acquired using an Inverted LSM 880 Airyscan NLO laser scanning confocal and multi-photon microscope (Zeiss). Either a 10 x/0.45 NA objective, or an immersion-corrected 25 x/0.8 NA or 63 x/1.4 NA objective was used at a resolution of 1024x1024 pixels. If tiling was used, images were stitched with 10% or 12% overlap. Laser power, gain, and other parameters were individually optimized to acquire highest quality images for ovary samples from non-blood-fed and post-egg-laying animals. Confocal images were viewed and processed using FIJI/ImageJ, and single slices were selected as representative images.

### Identification of orthologs

Orthologs for *tweedledee* and *tweedledum* in *Aedes albopictus* were identified using orthology relationships in VectorBase. We noted that the current release of the *Aedes albopictus* genome assembly (GCF_006496715.1 as of December 2021) contained multiples copies of the locus with *tweedledee*, *tweedledum*, *scratch* and *peritrophin-like*. With the currently available data, we were unable to ascertain whether the multiple copies of the locus reflect true duplication events, or incompletely collapsed haplotypes. We arbitrarily chose one locus for subsequent analyses with *Aedes albopictus* genes, but repeating analyses with genes in a second locus yielded no significant differences. Putative orthologs in *Culex quinquefasciatus* and all specified *Anopheles* species were found by searching syntenic genomic regions between annotated orthologs of *peritrophin-like* and *scratch* on VectorBase. *Anopheles* genome annotations were used for further analyses if *peritrophin-like, scratch* exon 1 and the *putative orthologs* were all unambiguously annotated, and if all three genes were found on the same contig. *Aedes* and *Culex* protein sequences were obtained from NCBI. *Anopheles* protein sequences were also downloaded from NCBI if a RefSeq annotation was available, but if unavailable as with *Anopheles quadriannulatus* and *Anopheles culicifacies*, protein sequences were taken from VectorBase. Multiple sequence alignments and protein sequence identity matrices were generated using MUSCLE (*Madeira et al., 2019*). The annotation of *scratch* was fragmented, with both exons annotated as separate genes in *Aedes aegypti*, *Anopheles arabiensis*, *Anopheles culicifacies* and *Anopheles quadriannulatus*, likely due to the presence of a large >50 kb intron. To minimize the ambiguity of the *scratch* protein sequences in these species, we generated the multiple sequence alignment between only the first exon of *scratch* in each species. Gene accession numbers are available at https://doi.org/10.5281/zenodo.5945524.

### Guanine+cytosine (GC) content analysis

GC content for all protein-coding genes from *Aedes aegypti* (AaegL5) was retrieved from Ensembl Metazoa BioMart (version 0.7) (*Kinsella et al., 2011*) using VectorBase as the gene source. The search was then limited to genes with a predicted cleavage site (SignalP 4.1) to filter for protein-coding genes with a predicted signal peptide.

## Amino acid content analysis

All protein sequences with a predicted signal peptide encoded by the *Aedes aegypti* (AaegL5) genome were retrieved from Ensembl Metazoa BioMart (version 0.7; *Kinsella et al., 2011*). The signal peptide predicted was cleaved for each protein using SignalP 4.1 (*Nielsen, 2017*), and the percent of each amino acid was then calculated for the cleaved sequence. Mean percent residue was calculated for 3,040 *Aedes aegypti* proteins (minimum protein length = 60 amino acids) with predicted signal peptides.

## dN/dS ratio

We aligned coding sequences of 8,030 protein-coding *Aedes aegypti* genes to unique orthologs in *Aedes albopictus*, and coding sequences of 9958 protein-coding *Anopheles gambiae* genes to unique orthologs in *Anopheles stephensi*, as annotated in Ensembl Metazoa, via PRANK (*Löytynoja, 2014*) using the codon option. dN/dS values per gene were calculated with KaKs_calculator (*Wang et al., 2010*) using the YN model (*Yang and Nielsen, 2000*). Sliding window values of dN/dS for *Aedes tweedledee* and *tweedledum* were calculated using a custom script for KaKs_calculator available at https://github.com/LiZhaoLab/Kaks_Calculator, (*Zhao, 2021*)(copy archived at swh:1:rev:4a40862b-c3e436262a4367157503ebb8a6e7f7f5). For comparing dN/dS ratios of ovary-expressed genes, we filtered results to include 6244 protein-coding *Aedes aegypti* genes with TPM >2 in bulk ovary RNA-seq from at least one reproductive time point.

## Population genetics analysis of whole-genome sequencing data

Paired fastq files from whole-genome sequencing of individual *Aedes aegypti* mosquitoes were downloaded from SRA (Accession: SRP246931) (*Rose et al., 2020*). After excluding data from populations with fewer than 8 sequenced individuals or those sampled outside sub-Saharan Africa, we retained 407 individuals from 25 African populations for further analysis. The fastq files were aligned to the *Aedes aegypti* L5 reference genome (*Matthews et al., 2018*) using BWA-MEM (*Li, 2013*) with default parameters. Chromosomal coordinates for each protein-coding gene were defined using AaegL5 RefSeq gene start to gene end defined as spanning the 5' and 3' extent of the mRNA and any intervening introns, and multiallelic genetic variants in these loci were called using bcftools ('bcftools mpileup | bcftools call -mv | bcftools norm -m -') (*Danecek et al., 2021*). Variants with missing genotypes in over 90% of individuals in any population and variant quality below 30 ('bcftools view -e 'F_MISSING >0.9' | rtg vcffilter -q 30') were excluded (*Danecek et al., 2021*; *Cleary et al., 2015*). Missense variants were annotated using SnpEff (*Cingolani et al., 2012*). Population allele frequency for each variant was calculated using bcftools (*Danecek et al., 2021*; 'bcftools query -f '%CHROM %POS %REF %ALT %AN %AC{0}\n''). Finally, alternate variants with allele frequency less than 5% in all populations were removed.

## McDonald-Kreitman (MK) test

To run an MK test for *tweedledee* and *tweedledum* in *Aedes aegypti*, we used population genetics data processed as described above and additionally included 47 sequenced individuals sampled outside sub-Saharan Africa, for a total of 454 genomes. After variants were called, we generated alternative coding sequences of each gene using SNPs in each individual and realigned using PRANK with the -codon function (*Löytynoja and Goldman, 2008*) with the orthologous gene in *Aedes albopictus*. To run the MK test for the *putative ortholog* in *Anopheles gambiae*, we downloaded VCF files for 1539 individuals from the Ag1000G project (*The Anopheles gambiae 1000 Genomes Consortium, 2021*). We generated alternative coding sequences using SNPs in these individuals and realigned using PRANK -codon using the orthologous gene in *Anopheles stephensi*. For each gene, we estimated the proportion of substitutions driven by positive selection (alpha) and used a Fisher's Exact test to calculate the p-value.

## Correlation to ecological variables

Human population density, host preference index, and geographic coordinates for each *Aedes aegypti* population's sampling location were acquired from *Rose et al., 2020*, Table S1 (56). WorldClim Version 2 was used to query 19 Bioclimatic variables using a spatial resolution of 30 s (*Fick and Hijmans, 2017*). We calculated Pearson's correlation between each of the 21 ecological variables (19 Bioclim variables

+ human population density + host preference index, the two anthropological variables) and population allele frequency in each sampling location, for all genetic variants called in protein-coding genes. For illustration purposes, a subset showing 12 temperature and precipitation-related Bioclim variables are shown along with the anthropological variables in *Figure 5*. The remaining data are available on Zenodo (https://doi.org/10.5281/zenodo.5945524). The corresponding p-value for the correlation between each ecological variable and each genetic variant was calculated using a two-tailed Pearson's correlation test. For each protein-coding gene, we computed the combined p-value for overall correlation to each ecological variable using the harmonic mean p-value test to combine individual p-values of all genetic variants and correct for multiple testing. We ranked combined p-values for all protein-coding genes for all ecological variables and calculated the percentile distribution for each of them. Correlation test is plotted in *Figure 5C* as the $-\log_{10}$(harmonic mean combined p-value).

## Permutation tests for *tweedledee*, *tweedledum*, and *Or4*

To determine whether the observed correlations with climate variables for *tweedledee*, *tweedledum* and *Or4* are greater than expected, we ran permutation tests for each gene and variable. We randomly sampled genetic variants from all protein-coding genes to simulate 10,000 genes with the same number of genetic variants as either *tweedledee* (358), *tweedledum* (292) or *Or4* (1089). Significance was defined as p<0.05, that is, if *tweedledee*, *tweedledum* or *Or4*, respectively, had mean gene-wide correlation greater than 9500 simulated genes.

## Generation of Δ*deedum* double mutants

The Δ*deedum* double mutant was generated using CRISPR-Cas9 (*Kistler et al., 2015*). Wild type embryos of the *Aedes aegypti* Liverpool strain were injected at the Insect Transformation Facility at the University of Maryland Institute for Bioscience and Biotechnology Research with a gene-targeting mixture composed of 300 ng/µL Cas9 protein with NLS (PNA Bio, CP01-200) and 4 sgRNAs, each 40 ng/µL. Two of the sgRNAs targeted exon 2 of *tweedledee* and the other two targeted exon 2 of *tweedledum* (see https://doi.org/10.5281/zenodo.5945524). Coordinates on *Aedes aegypti* chromosome 2 for *tweedledee*: 113,795,266–113,794,685 and *tweedledum:* 113,807,172–113,806,119, as annotated in the AaegL5 genome (*Matthews et al., 2018*). As described (*Kistler et al., 2015*), DNA templates were generated for each sgRNA by annealing oligonucleotides using the NEBNext Ultra II Q5 master mix (NEB, M0544L). The HiScribe Quick T7 kit (NEB, E2050S) was then used for in vitro transcription, per manufacturer's instructions, with an overnight incubation of 17 hr at 37 °C. Prior to mixing with Cas9 protein, sgRNAs were purified using SPRI beads (Beckman-Coulter, Ampure RNAclean, A63987) with elution in Ultrapure DNase/RNase-free distilled water (Invitrogen, 10977–015). The mutant allele was identified using polymerase chain reaction (PCR) and confirmed to be a double mutant, Δ*deedum,* in which both *tweedledee* and *tweedledum* were disrupted. The strain was backcrossed to wild type Liverpool animals for a minimum of four generations before inbreeding to homozygose. The homozygous Δ*deedum* strain was successfully established and behaviorally phenotyped. To verify wild type, +/Δ*deedum*, and Δ*deedum*/Δ*deedum* animals, three independent PCRs were run on each of the DNA template genotypes as described in *Figure 6—figure supplement 1* and *Figure 6—figure supplement 1—source data 1*. The corresponding genotyping primers are listed at https://doi.org/10.5281/zenodo.5945524.

## Generation of Δ*dum* single mutant and attempted generation of Δ*dee* single mutant

Δ*dum*, a *tweedledum* single mutant that was wild type at the *tweedledee* locus was recovered using the same mutagenesis procedure as the Δ*deedum* double mutant described above. The same genotyping strategy as that used for the Δ*deedum* double mutant was used with the Δ*dum* single mutant to confirm that only *tweedledum* was mutated. With a distinct cocktail of sgRNAs, an allele with a deletion in *tweedledee* (Δ*dee*) that spared *tweedledum* was also isolated but attempts to homozygose and establish a Δ*dee* mutant have been unsuccessful to date.

## Photographs of ovaries and spermathecae

Mosquitoes were cold-anesthetized and kept on ice for up to 1 hr, or until dissections were complete. Ovaries and spermathecae were dissected on ice in 1 X PBS. Ovaries were photographed using an

AxioCam ERc 5 s camera (Zeiss) attached to a stereo microscope (Zeiss, SteREO Discovery KMAT). Spermathecae were photographed using an iPhone X through the iDu Optics LabCam adapter attached to the eyepiece of a wide-field compound microscope (Swift, SW350B).

## Statistical analysis

R (version 4.1.1) and GraphPad Prism 9 software were used for data visualization and statistical analysis except when specified otherwise in the sections above or in the figure legends.Data and resource availability

## Data and resource availability

All raw data reported here, along with a TPM count table from ovary RNA-seq, two protein abundance (iBAQ) tables from hemolymph and ovary proteomics respectively, population genetic analysis, and instructions for fabricating and using the blood puck are available on Zenodo (https://doi.org/10.5281/zenodo.5945524).

## Acknowledgements

We thank members of the Vosshall Lab for comments on the manuscript; Gloria Gordon and Libby Mejia for expert mosquito rearing; James Petrillo at The Rockefeller University Precision Instrumentation Technologies Resource Center for input on design, optimization, and fabrication of the metal blood puck; Vadim Sherman at the Rockefeller High-Energy Physics Machine Shop for fabrication of the metal blood puck; Rob A Harrell II at the Insect Transgenesis Facility at the University of Maryland for CRISPR-Cas9 embryo injections; Benjamin J Matthews for advice on the design of sgRNAs and egg-laying assays; Laura B Duvall for advice on the design of host-seeking suppression behavioral assays; Katarzyna Cialowicz, Christina Pyrgaki, Carlos Rico, and Alison North at The Rockefeller University Bio-Imaging Resource Center for training and advice on confocal imaging (RRID:SCR_017791); Leah Houri-Zeevi for advice on RNA-seq experiment design, optimization, and data analysis; Thomas Carroll and Douglas Barrows for assistance with RNA-seq data analysis; Connie Zhao at The Rockefeller University Genomics Resource Center for assistance with RNA-seq library preparation and sequencing; Caroline Jiang for advice on statistical analysis of behavior data; Junhui Peng for discussion on protein structure predictions; Alexander Wild for mosquito photographs; Daniel Kronauer and Shai Shaham for general discussion and project guidance.

Funding for this study was provided by the Boehringer Ingelheim Fonds PhD fellowship (KV); a predoctoral fellowship from the Kavli Neural Systems Institute and NIH NIDCD grant F30DC017658 (MH); EMBO ALTF 286–2019 (NS); NIH MIRA R35GM133780, Robertson Foundation, Monique Weill-Caulier Career Scientist Award, Rita Allen Foundation Scholar Program, and a Vallee Scholar Program (VS-2020–35) (LZ); Fellowship of Tsinghua Xuetang Life Science Program (J Zhao); NRSA Training Grant #GM066699 (LAN). The Proteomics Resource Center at The Rockefeller University acknowledges funding for mass spectrometer instrumentation from the Sohn Conferences Foundation and the Leona M and Harry B Helmsley Charitable Trust. LBV is an officer of the Howard Hughes Medical Institute.

## Additional information

### Funding

| Funder | Grant reference number | Author |
| --- | --- | --- |
| Boehringer Ingelheim Fonds | BIF PhD Fellowship | Krithika Venkataraman |
| Kavli Foundation | KNSI Pre-doctoral fellowship | Margaret Herre |
| National Institutes of Health | F30DC017658 | Margaret Herre |
| European Molecular Biology Organization | EMBO ALTF 286-2019 | Nadav Shai |

| Funder | Grant reference number | Author |
|---|---|---|
| National Institutes of Health | MIRA R35GM133780 | Li Zhao |
| Rita Allen Foundation | Rita Allen Scholar | Li Zhao |
| Vallee Foundation | VS-2020-35 | Li Zhao |
| National Institutes of Health | NRSA Training Grant #GM066699 | Lauren A Neal |
| Leona M. and Harry B. Helmsley Charitable Trust | | Henrik Molina |
| Sohn Conferences Foundation | | Henrik Molina |
| Monique Weill-Caulier Career Scientist Award | | Li Zhao |
| Robertson Foundation | | Li Zhao |
| Howard Hughes Medical Institute | | Leslie B Vosshall |
| Fellowship of Tsinghua Xuetang Life Science Program | | Jieqing Zhao |

The funders had no role in study design, data collection and interpretation, or the decision to submit the work for publication.

## Author contributions

Krithika Venkataraman, Conceptualization, Data curation, Formal analysis, Supervision, Funding acquisition, Validation, Investigation, Visualization, Methodology, Writing - original draft, Project administration, Writing - review and editing; Nadav Shai, Conceptualization, Validation, Investigation, Visualization, Methodology; Priyanka Lakhiani, Conceptualization, Formal analysis, Visualization, Methodology; Sarah Zylka, Investigation, Methodology; Jieqing Zhao, Joshua Zeng, Lauren A Neal, Investigation; Margaret Herre, Data curation, Formal analysis, Investigation, Methodology; Henrik Molina, Formal analysis, Investigation, Methodology; Li Zhao, Conceptualization, Software, Supervision, Project administration; Leslie B Vosshall, Conceptualization, Resources, Supervision, Funding acquisition, Project administration, Writing - review and editing

## Author ORCIDs

Krithika Venkataraman  http://orcid.org/0000-0002-2067-2387
Nadav Shai  http://orcid.org/0000-0002-2812-3884
Priyanka Lakhiani  http://orcid.org/0000-0003-2797-8650
Sarah Zylka  http://orcid.org/0000-0002-7311-2981
Jieqing Zhao  http://orcid.org/0000-0002-0134-7538
Margaret Herre  http://orcid.org/0000-0001-7868-3321
Joshua Zeng  http://orcid.org/0000-0002-4694-3309
Lauren A Neal  http://orcid.org/0000-0002-0092-2852
Henrik Molina  http://orcid.org/0000-0001-8950-4990
Li Zhao  http://orcid.org/0000-0001-6776-1996
Leslie B Vosshall  http://orcid.org/0000-0002-6060-8099

## Ethics

Behavioral experiments and blood-feeding using live hosts were approved and monitored by The Rockefeller University Institutional Review Board (IRB protocol LV-0652). All human subjects gave their written informed consent to participate in this study.
Blood-feeding using live mice was approved and monitored by The Rockefeller University Institutional Animal Care and Use Committee (IACUC protocol 17018).

## Decision letter and Author response

Decision letter https://doi.org/10.7554/eLife.80489.sa1
Author response https://doi.org/10.7554/eLife.80489.sa2

# Additional files

## Supplementary files
• MDAR checklist

## Data availability

RNA-sequencing data have been deposited in GEO under accession code GSE193470. The mass spectrometry proteomics data have been deposited to the ProteomeXchange Consortium via the PRIDE partner repository with the dataset identifier PXD030925. Ovary sample raw files begin with the code "MS205850LUM". Hemolymph sample raw files begin with the code "MS195106LUM". All raw data reported here, along with a TPM count table from ovary RNA-seq, two protein abundance (iBAQ) tables from hemolymph and ovary proteomics respectively, population genetic analysis, and instructions for fabricating and using the blood puck are available on Zenodo: https://doi.org/10. 5281/zenodo.5945524.

The following datasets were generated:

| Author(s) | Year | Dataset title | Dataset URL | Database and Identifier |
|---|---|---|---|---|
| Venkataraman K, Vosshall LB, Shai N | 2022 | *Aedes aegypti* ovary bulk RNA-seq | https://www.ncbi. nlm.nih.gov/geo/ query.acc.cgi?acc= GSE193470 | NCBI Gene Expression Omnibus, GSE193470 |
| Venkataraman K, Vosshall LB, Molina H, Zhao J, Herre M | 2022 | *Aedes aegypti* ovary and hemolymph proteomes | http:// proteomecentral. proteomexchange. org/cgi/GetDataset? ID=PXD030925 | ProteomeXchange, PXD030925 |
| Venkataraman K, Shai N, Lakhiani P, Zylka S, Zhao J, Herre M, Zeng J, Neal LA, Molina H, Zhao L, Vosshall LB | 2023 | Two novel, tightly linked, and rapidly evolving genes underlie Aedes aegypti mosquito reproductive resilience during drought | https://doi.org/10. 5281/zenodo.5945524 | Zenodo, 10.5281/ zenodo.5945524 |

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
