## [Editor Report]

This important study focuses on egg retention in the face of desiccating conditions in the mosquito *Aedes aegypti*. The research identifies genes associated with a trait that could be important to explain the vectorial capability of *Aedes aegypti* to transmit disease and expand into a cosmopolitan range. The presented evidence is convincing and the implications are well-articulated. The results should be of importance for ecological geneticists and vector biologists alike.

---

## [Decision Letter]

**Decision letter after peer review:**

Thank you for submitting your article "Rapidly evolving genes underlie Aedes aegypti mosquito reproductive resilience during drought" for consideration by *eLife*, and sorry for the very long time it took to obtain the reviews and for the reviewers to discuss their critiques. Your article has been reviewed by 2 peer reviewers, and the evaluation has been overseen by a Reviewing Editor and Claude Desplan as the Senior Editor. The reviewers have opted to remain anonymous.

Even though we all saw potential in the research, the manuscript needs substantial changes before we can consider it for publication in *eLife*. The reviewers have discussed their reviews with one another, and the Reviewing Editor has drafted this to help you prepare a revised submission.

The study focuses on egg retention in the face of desiccating conditions in the mosquito Aedes aegypti. The research identifies genes associated with a trait that could be important to explain the vectorial capability of Ae. aegypti to transmit disease and expand into a cosmopolitan range. Given the truly gigantic genome size of the Aedes genome, the authors resort to the use of a differential expression and candidate gene approach which the authors leverage to generate genetic mutants in mosquitoes. This eventually leads to the identification of two loci that might be associated with the trait. The reviewers found the differentiation expression approach somehow limited because genes with low levels of expression might also be important to explain a sizable portion of the genetic variance of the trait. This possibility does not appear mentioned in the manuscript.

The second component of the manuscript is the generation of genetic knockouts. This section seems technically sound and the research group has led the efforts on this matter. Nonetheless, and given the differential expression approach, these findings are preliminary because the piece does not identify the architecture of the trait. This concern is somehow, but not completely, alleviated by the results on panel 5I.

This leads to a third concern: The figures are inscrutable. Part of the delay to finalize the review of this manuscript was the complex presentation of the results which took the reviewers a long time to digest. Any revised version of this manuscript will require a major overhaul of the presentation of the results.

The final concern is the lack of scholarship in the manuscript. There are several references that should be cited that do not appear in the text. This issue is particularly apparent in the area of evolutionary biology. The work is not contextualized in the larger picture of copy number evolution, gene duplication, and genetic architecture broadly defined. The authors for example coined the term 'conceptualog' which is simply a potential homolog. Coining terms like this one is not helpful and just creates confusion in the literature. The reviewers offer useful suggestions but the authors will need to make an effort to better provide the readers with the tools to understand the importance of their work.

Essential revisions:

1. Consider the caveats of the differential gene expression approach. Elaborate on how these limitations affect your conclusions. It is particularly important to discuss the findings in the context of the potential genetic architecture of the trait.

2. Make the figures and legends more accessible to the reader.

3. Improve the scholarship of the piece. Review the body of work on gene duplication and the origin of new genes.

4. Remove the concept of 'conceptualog'.

*Reviewer #1 (Recommendations for the authors):*

1. Please do not coin any new terms such as "conceptualog". It is not required and is unnecessary. One can use a simple term such as a "potential homolog" or "putative homolog" or "putative orthologs".

*Reviewer #2 (Recommendations for the authors):*

I overall think this is a very strong manuscript. I only have one major suggestion.

I do recommend that in addition to looking at the top 50 expressed genes during egg retention, I believe the authors should use an approach like WGCNA (Zhang and Horvath, 2005) to look for gene modules that are upregulated during egg retention and down-regulated otherwise, and examined a broader list of candidate genes. This approach should allow the authors to strengthen two conclusions. First, they can examine the frequency of taxon-restricted genes in such modules vs. random expectation (of ovary-expressed genes in general). Second, this should allow the authors to compare dN/dS not to the overall genome, but to other ovary-limited genes which in general seem to show rapid evolutionary rates in mosquitoes. As I realize this may be more analyses than the authors wish to perform, I think the alternative is to simply acknowledge that their candidate gene approach has biased them to look specifically at taxon-restricted genes and to identify genes with potentially higher evolutionary rates as caveats in their discussion.

---

## [Author Response]

Essential revisions:1. Consider the caveats of the differential gene expression approach. Elaborate on how these limitations affect your conclusions. It is particularly important to discuss the findings in the context of the potential genetic architecture of the trait.

We have revised the manuscript to make it clear that we used a candidate gene approach to study the functions of two genes. We also clarify that the focus of the manuscript is not on defining the genetic architecture of a trait and is instead on the discovery of two novel genes that are highly regulated and that contribute to an important phenotype, which enables female mosquitoes to remain reproductively resilient even in the face of drought stress.

2. Make the figures and legends more accessible to the reader.

We have modified our narrative of the results, specifically for Figure 2, to make the figures simpler to understand. We have also simplified Figure 2 such that it describes the abundant expression of these two candidate genes, and provides evidence for their precisely controlled temporal- and tissue-restricted expression. This highlights the uniqueness of these genes and their compelling characteristics, focusing on these reasons as reasons for further studying them, rather than the use of a differential gene expression approach with limitations and caveats.

3. Improve the scholarship of the piece. Review the body of work on gene duplication and the origin of new genes.

We have revised the manuscript to make our scope and emphasis clear and have also included more references to literature in the field of evolution.

4. Remove the concept of 'conceptualog'.

We have removed “conceptualog” and changed the word to “putative ortholog” throughout the paper and figures.

Reviewer #1 (Recommendations for the authors):1. Please do not coin any new terms such as "conceptualog". It is not required and is unnecessary. One can use a simple term such as a "potential homolog" or "putative homolog" or "putative orthologs".

At the Reviewer’s request, we have removed the term from the text and the figures and replaced it with “putative ortholog.”

Reviewer #2 (Recommendations for the authors):I overall think this is a very strong manuscript. I only have one major suggestion.I do recommend that in addition to looking at the top 50 expressed genes during egg retention, I believe the authors should use an approach like WGCNA (Zhang and Horvath, 2005) to look for gene modules that are upregulated during egg retention and down-regulated otherwise, and examined a broader list of candidate genes. This approach should allow the authors to strengthen two conclusions. First, they can examine the frequency of taxon-restricted genes in such modules vs. random expectation (of ovary-expressed genes in general). Second, this should allow the authors to compare dN/dS not to the overall genome, but to other ovary-limited genes which in general seem to show rapid evolutionary rates in mosquitoes. As I realize this may be more analyses than the authors wish to perform, I think the alternative is to simply acknowledge that their candidate gene approach has biased them to look specifically at taxon-restricted genes and to identify genes with potentially higher evolutionary rates as caveats in their discussion.

Thank you for your comments and suggestions. We appreciate your positive evaluation and constructive criticism. The manuscript was not meant to prove that all rapidly evolving genes are functionally important, but rather some (here in this paper, two) rapidly evolving genes can be important for evolution and adaptation. In the new version, we edited the manuscript to make the above points clear. Although we appreciate the reviewer’s suggestion of performing co-expression analysis, we decided not to include them in the revised version, as that would deviate from the major points of the already lengthy manuscript. Instead, we took the reviewer’s suggestion and acknowledged that we used a candidate gene approach.

The suggestion of performing dN/dS not to the overall genome is an excellent point. We have revised the analysis to analyze all ovary-expressed genes. The percentiles of dN/dS for *tweedledee* and *tweedledum* are 98% and 94%, slightly more extreme compared to the results using all genes, suggesting that dN/dS of the two genes are indeed outliers compared to all ovary-expressed genes. In line with this thought, we also performed a McDonald-Kreitman test (MK test) for the genes and found that both genes are marginally significant or insignificant (*tweedledee* p = 0.068; *tweedledum* p = 0.048), despite that these are small genes and have low Pn values.